# XFinBench: Benchmarking LLMs in Complex Financial Problem Solving and Reasoning

## Abstract

Solving financial problems demands complex reasoning, multimodal data processing, and a broad technical understanding, presenting unique challenges for current large language models (LLMs). We introduce **XFinBench**, a novel benchmark designed to evaluate LLM's ability in solving comple**X**, knowledge-intensive **Fin**ancial problems across diverse graduate-level topics with multi-modal context. We identify five core capabilities of LLMs using XFinBench, *i.e*, *terminology understanding*, *temporal reasoning*, *future forecasting*, *scenario planning*, and *numerical modelling*. XFinBench features 4,235 examples derived from graduate-level finance textbooks, and consists of three tasks: Statement Judging, Multi-choice Question Answering and Financial Calculation. Upon XFinBench, we conduct extensive experiments on 18 leading models. The result shows that o1 is the best-performing text-only model with an overall accuracy of 67.3%, but still lags significantly behind human experts with 12.5%, especially in *temporal reasoning* and *scenario planning* capabilities. We further construct a knowledge bank with 3,032 finance terms for knowledge augmentation analysis, and find that relevant knowledge to the question only brings consistent accuracy improvements across five capabilities to small open-source model. Additionally, our error analysis reveals that rounding errors in middle of calculation and blindness to position and intersection of curves in the image are two primary issues leading to model's poor performance in calculating and visual-context questions, respectively. These findings underscores the critical role XFinBench will play in the development of general-purpose of AI agents of tackling complex, knowledge-intensive financial problems with multi-modal context.

## 1 Introduction

Finance constitutes a critical domain, characterized by the necessity for sophisticated problem-solving skills. Beyond domain-specific knowledge, it necessitates advanced capabilities such as temporal reasoning (Su et al., 2024; Wang & Zhao, 2024), future forecasting (Jin et al., 2024; Zhou et al., 2023b), scenario planning (Valmeekam et al., 2022; Geva et al., 2021), and numerical modeling (Zhao et al., 2024; Koncel-Kedziorski et al., 2024). Besides, complex finance problems in real world usually involves rich multimodal information, covering time series (Yu et al., 2023), long tabular (Reddy et al., 2024) and various charts (Masry et al., 2022; Lu et al., 2024). These complexities present significant challenges for large language models (LLMs), thereby rendering finance an appropriate testbed for the evaluation of LLMs.

Numerous datasets have been curated to assess the reasoning abilities of AI systems in the finance domain, with most emphasizing quantity extraction and basic mathematical reasoning (see Table 1). Existing datasets, including TAT-QA (Zhu et al., 2021), FinQA (Chen et al., 2021), MultiHiertt (Zhao et al., 2022), PACIFIC (Deng et al., 2022) and ConvFinQA (Chen et al., 2022), primarily focus on quantity extraction and basic numerical calculations using company's financial reports. However, they lack questions that entail extensive financial knowledge or complex reasoning processes. More recently, some benchmarks have been introduced to assess the performance of LLMs on knowledge-intensive finance tasks. For instance, BizBench (Koncel-Kedziorski et al., 2024) collects past finance datasets for quantity extraction and knowledge examination to test LLMs' business and financial understanding; KnowledgeFMATH (Zhao et al., 2024) emphasize LLMs' mathematical reasoning and code completion abilities within the finance domain; and FinEval (Zhang et al., 2023) focuses on

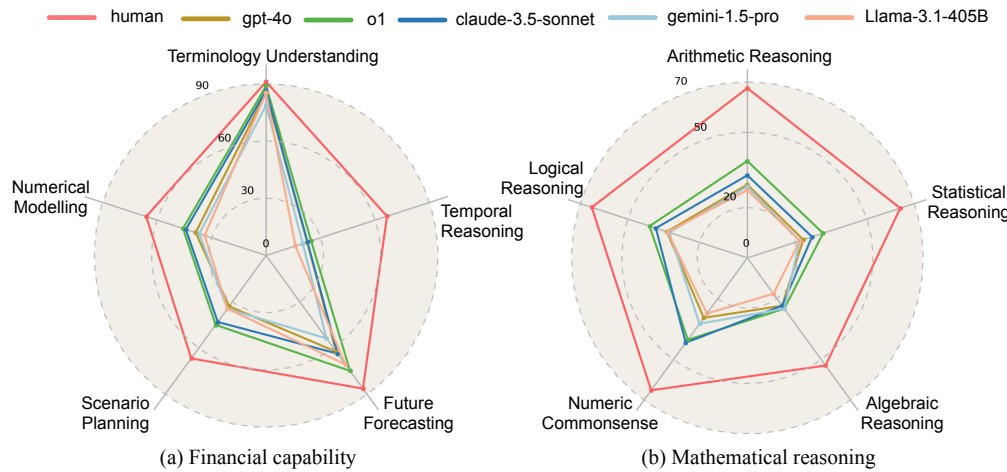

Figure 1: Accuracies of leading LLMs and human performance on XFINBENCH across (a) five capabilities for complex finance problem solving and (b) mathematical reasoning types(Lu et al., 2024). Accuracies for o1 and Llama-3.1-405B here do not include questions with visual context.

understanding finance concepts in Chinese. Nevertheless, these benchmarks still do not address the advanced capabilities necessary for solving complex financial problems like temporal reasoning, forecasting, and planning. To bridge this gap, we introduce XFINBENCH, a novel benchmark specifically designed to evaluate LLM's ability in solving **complex**, **knowledge-intensive financial problems** across diverse graduate-level topics with multi-modal context.

XFINBENCH consists of 4,235 examples derived from graduate-level finance textbooks that ensures the complexity of financial problems in our dataset, and brings convenience to annotation of ground-truth knowledge to each problem. Different from existing datasets that only evaluate the model's grasp of specialized financial vocabulary, *i.e, Terminology Understanding*, XFINBENCH identifies four more advanced capabilities essential for complex finance problem-solving: (1) *Temporal Reasoning*, involving the comprehension of time-based data and temporal relationships; (2) *Future Forecasting*, testing logical reasoning in predicting financial trends based on theoretical finance models; (3) *Scenario Planning*, analyzing different potential future scenarios to assess their impact on financial decisions and strategies; and (4) *Numerical Modelling*, which involves constructing structured representations of companies and products' financial performance. Moreover, XFINBENCH includes three tasks: *statement judging*, which evaluates the model's understanding of finance concepts; *multi-choice question answering*, which assesses strategic decision-making and predictive capabilities with visual data; and *financial calculation*, which tests mathematical reasoning in finance. To further investigate how domain-specific knowledge could boost LLM's performance on our complex financial problems, we also design a knowledge bank with 3,032 finance terms and ask human experts to annotate the ground-truth knowledge to each problem. Detailed capability definitions can be found in §A, and data collection pipeline in §2 and §A.

We conduct extensive experiments on XFINBENCH to evaluate the complex finance problem-solving ability of 18 leading LLMs , along with knowledge augmentation analysis and error analysis. Our models include nine close-source models (*e.g*, o1, gpt-4o, claude-3.5-sonnet, etc.), two multi-modal open-source models (*i.e*, Llama-3.2-Vision 11B and 90B), and seven text-only open-source models (*e.g*, Llama-3.1, Mixtral 8×7B). We implement the Chain-of-Thought (CoT) prompting method for all three tasks, and additionally apply Program-of-Thought (PoT) for *financial calculation*. Moreover, we establish a human performance baseline of human experts with finance degree. We show that XFINBENCH, featuring graduate-level topics and advanced capabilities for complex finance problem-solving, is a challenging benchmark with human performance reaching only 79.8%.

Our results indicate that o1 is the best-performing text-only model with an overall accuracy of 67.3%, while claude-3.5-sonnet achieves the highest accuracy of 64.0% when visual-context questions included (§3.2). Despite that LLMs achieve comparable performance with human in *termi-*

Table 1: Comparison of XFINBENCH with existing datasets.

| Dataset | Size | Modality | Knowledge-intensive | Mathematical-Reasoning | Complex-Problem | Source |
|---|---|---|---|---|---|---|
| TAT-QA | 16,552 | Tabular | ✗ | ✓ | ✗ | Financial Report w. CrowdSource |
| PACIFIC | 2,757 | Tabular | ✗ | ✓ | ✗ | Existing dataset w. Automatic Pipeline |
| FinQA | 8,281 | Tabular | ✗ | ✓ | ✗ | Financial Report w. CrowdSource |
| ConvFinQA | 3,892 | Tabular | ✗ | ✓ | ✗ | Existing dataset w. CrowdSource |
| FinEval | 4,661 | None | ✓ | ✗ | ✗ | Chinese Textbook |
| BizBench | 19,842 | Tabular | ✓ | ✓ | ✗ | Existing Dataset, Certificate Exams |
| KnowledgeFMATH | 1,259 | Tabular | ✓ | ✓ | Partial | Internet w. CrowdSource |
| XFINBENCH | 4,235 | Tabular, Figure | ✓ | ✓ | ✓ | Graduate-level English Textbook w. CrowdSource and GPT-4o |

*nology understanding*, as shown in Figure 1, they still significantly lag behind human experts in more advanced capabilities for complex finance problem-solving, including *temporal reasoning* and *scenario planning*—especially when visual context is involved. Even if we augment models with ground-truth knowledge from knowledge bank, the improvements across advanced capabilities are still limited and inconsistent, except for small open-source model (§3.3). Moreover, our error analysis reveals that rounding error in the intermediate steps of calculation and model's blindness of position and intersection of curves in the image (Rahmanzadehgervi et al., 2024) are two inescapable issues leading to the poor performance in calculating and visual-context questions, respectively (§3.4). These findings highlight that XFINBENCH represents a rigorous and challenging benchmark, offering a critical tool for advancing the development of LLMs in complex financial problem-solving and reasoning.

## 2 DATASET CONSTRUCTION

Our benchmark, XFINBENCH, is developed to support complex reasoning in knowledge-intensive finance tasks. We began by collecting questions and answers from three graduate-level finance textbooks and their solution manuals, while also building a knowledge bank of finance terms. Human experts annotated each question-answer pair with relevant finance terms to enrich the dataset. However, since open-ended and calculation-based questions pose challenges for LLM evaluation, we leveraged GPT-4o to further annotate and expand the dataset, enhancing both its size and suitability for LLM assessments. Lastly, we conducted a rigorous quality validation process with human experts to ensure the dataset's accuracy and relevance. The final XFINBENCH dataset encompasses three key tasks—*statement judging*, *multi-choice question answering*, and *financial calculation*—and is supplemented by a comprehensive knowledge bank of finance terms and definitions.

### 2.1 INITIAL DATA COLLECTION

**Collection of Initial QA datasets.** To ensure the complex and knowledge-intensive properties of our benchmark, we extract after-class questions from three classic graduate-level finance textbooks that cover most finance topics, i.e. *Fundamentals of Corporate Finance*, *Options Futures and Other Derivative*, and *The Economics of Money Banking and Financial Markets*. We also download their solution manuals from official websites to collect the gold answers to their after-class questions. We then leverage OCR techniques via `pdfplumber` library to extract the text from PDF of textbooks and solution manuals. We extract the questions and answers at the end of each chapter, and take screenshots of tables and figures in context if any. In total, we collect 2,018 after-class questions from textbooks, 343 of them with visual or tabular context. Tabular context saved in images are processed by GPT-4o-mini to be stored in LATEX format.

**Classifying QA into Tasks.** We classify after-class questions collected from textbooks into three tasks: *statement judging*, *multi-choice question answering*, and *financial calculation*. Questions that evaluate the basic understanding of finance concepts and theoretical models are classified into *statement judging* task. Questions that focus on the application of financial strategies and models are classified into *multi-choice question answering* task. Some questions may be classified into both two tasks. For questions that involve numerical reasoning, we classify them into *financial calculation*

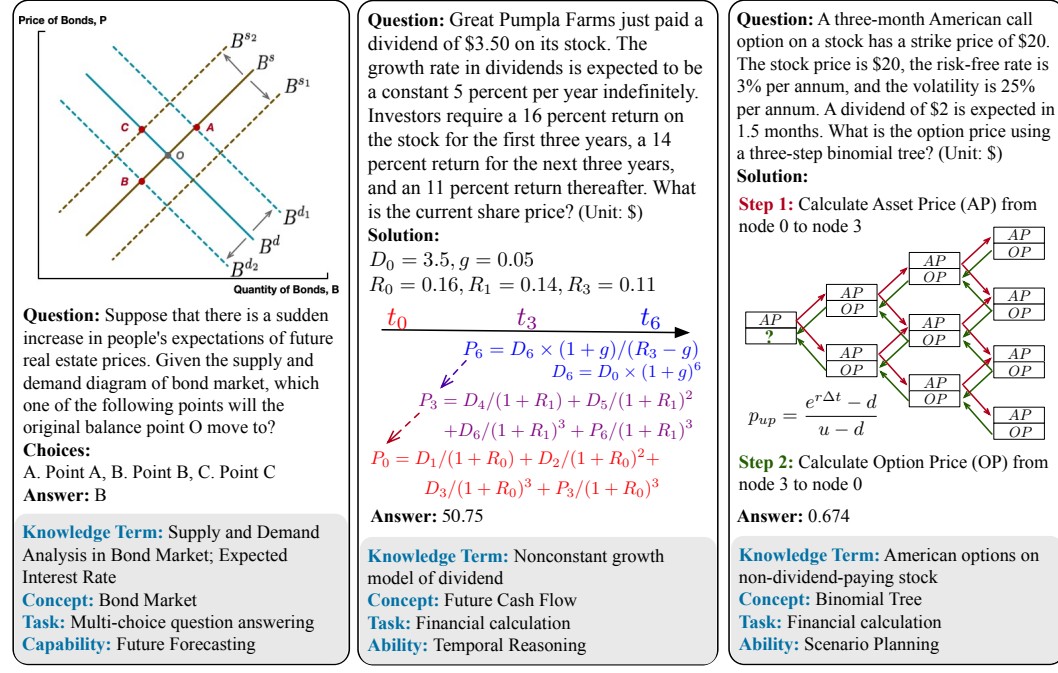

Figure 2: Examples in our dataset XFINBENCH.

task. Finally, 813 questions belong to the *statement judging* task, 624 to the *multi-choice question answering* task, and 858 to the *financial calculation* task.

**Collection of Knowledge Bank.** We construct a knowledge bank that consists of finance terms with definition for knowledge augmentation analysis during evaluation. We use the subject index at the end of each textbook to acquire the finance terms and their corresponding page ranges in textbook. We then manually extract the definition of each term from the corresponding pages. It is worth noting that some terms may share the same pages, indicating that they share the same definition. In total, we collect 3,032 terms with 1,766 unique definitions. Further details of knowledge bank construction can be found in §B.3.

**Bridging QA and Knowledge Bank.** We so far have collected after-class question-answer pairs and finance terms in each textbook, which are initially linked through chapters. In each chapter, a collection of finance terms is introduced in the main body, followed by after-class questions in the end. Human experts are then instructed to annotate each after-class question with 1-to-3 most relevant finance terms from the main body of the same chapter. Finally, a question is annotated with 1.3 terms on average. Further details of human annotation can be found in §B.3.

## 2.2 GPT-4O ENHANCED ANNOTATION

After-class questions from textbooks are mostly open-ended or consisting of a series of sub-questions, making it difficult to evaluate the model's response. For instance, the answer to the open-ended question "*Discuss the advantages and disadvantages of options and forward contracts*" includes a list of properties of options and future contracts; the calculation question "*An investment offers ... If the payment occurs for 15 years, what is its value? For 40 years? Forever?*" contains a series of sub-questions with different final answers. To ensure each question in XFINBENCH having a clear final answer to be evaluated accurately and conveniently, we leverage GPT-4o to process these questions under a Generate-then-verify framework (Zhang et al., 2024).

We first use few-shot prompts to ask GPT-4o to transform open-ended questions into those with clear final answers. For *statement judging* task, we ask GPT-4o to extract both true and false statements from each after-class question (see Appendix G.2.1). To ensure a balanced representation of true and

false statements, we apply two prompt templates with the same after-class questions as few shots, but one with true statements and one with false statements. For *multi-choice question answering* task, we follow STARC rules (Berzak et al., 2020) to ask GPT-4o to first extract a clear and complete question from the after-class question, and then create three candidate choices given the gold answer (see Appendix G.2.2). Among these choices, one is the correct answer with evidence, and the other two are misleading choices that either shows a misunderstanding of the gold answer or is made up by GPT-4o itself. For *financial calculation* task, we ask GPT-4o to split the after-class question into a series of independent questions with clear final answers (see Appendix G.2.3). In this stage, 6,227 questions are generated from after-class questions.

We then leverage GPT-4o to verify the quality of questions in the generation stage from multiple dimensions. We primarily evaluate *Correctness* and *Completeness* of the generated question and answer. Specifically, we evaluate whether (1) the question provides the *complete* background information to get its final answer, and (2) the final answer is *correct* to the question given the after-class question and its gold answer. Furthermore, to ensure the independence of questions in *statement judging* task, we verify if, within the same after-class question, true statements provide no evidence to support that false statement(s) is wrong. For *multi-choice question answering* task, we verify if the two misleading choices are exclusive to, but share the similar wording and length with the correct choice. For *financial calculation* task, we verify if the final answers are numerical without any text included. Finally, 35.2% questions are discarded in the verification stage. Details of automatic annotation can be found in §B.2.

### 2.3 DATA QUALITY VALIDATION

We conduct a comprehensive validation protocol to ensure the high quality of the annotated data. For each annotated question, we assign three evaluators to validate whether: 1) the question is fluent and contains complete information to get the final answer; 2) the final answer is correct according to the gold answer of after-class question; 3) the annotated finance terms are helpful for answering the question. We ask the evaluators to rate all examples in XFINBENCH on a scale of 1 to 5 individually. We then calculate the proportions of examples with average score S $\geq$ 4: *question fluency* 97.1%, *question completeness* 96.8%, *answer correctness* 98.0%, *knowledge helpfulness* 91.2%. The high scores illustrate the high quality of XFINBENCH. Further details can be found in §C.1.

### 2.4 DATA STATISTICS

The main statistics of XFINBENCH are presented in Table 2. XFINBENCH consists of 4,235 examples, divided into two subsets: *validation* and *test*. The division is based on random sampling over the after-class questions from textbooks. *validation* contains 1,000 examples, intended for model development validation or for those with limited computing resources. The test set features the remaining 3,235 examples for standard evaluation. Notably, the answer labels for *test* will not be publicly released to prevent data contamination, and we will maintain an online evaluation platform. Additionally, the knowledge bank consists of 3,032 finance terms with 1,766 unique definitions. There are 28 finance concepts in our benchmark, exceeding most existing datasets (see Figure 3). Detailed statistics of XFINBENCH and knowledge bank can be found in §C.

## 3 EXPERIMENTS

We conduct qualitative and quantitative studies to provide a comprehensive evaluation of leading LLMs for complex reasoning in knowledge-intensive finance tasks using XFINBENCH.

### 3.1 EXPERIMENTAL SETUP

We evaluate the models on the test set of XFINBENCH uder two setups: 1) *Multimodal Large Language Models* (MLLMs) who allow visual input, including gpt-4o (OpenAI, 2024b), gpt-4o-mini (OpenAI, 2024a), claude-3.5-sonnet (Anthropic, 2024a), claude-3-opus, claude-3-haiku (Anthropic, 2024b), gemini-1.5-flash and gemini-1.5 pro (Team, 2024b), and Llama-3.2-Vision models (Meta, 2024b), and 2) *Text-only Large Language Models* who only allow textual input, including o1 (OpenAI, 2024d), o1-mini (OpenAI, 2024c), deepseek-chat (DeepSeek-AI, 2024), Llama-3.1 models

Table 2: Key statistics of XFINBENCH.

| Statistics | Number |
|---|---|
| *XFINBENCH dataset* | |
| Total questions | 4,235 |
| - *statement judging* | 1,795 (42.4%) |
| - *multi-choice question answering* | 761 (18.0%) |
| - w. Image | 146 |
| - *financial calculation* | 1,679 (39.6%) |
| - w. Tabular | 330 |
| Question Length (Median / Avg) | 244 / 273.7 |
| Terms per question (Median / Avg) | 1.0 / 1.3 |
| Test Set Size | 3,235 |
| Validation Set Size | 1,000 |
| *Knowledge Bank* | |
| Total terms | 3,032 |
| Unique number of definition | 1,766 |
| - w. Mathematical Formula | 34.3% |
| Definition Length (Median / Avg) | 830 / 1,249 |

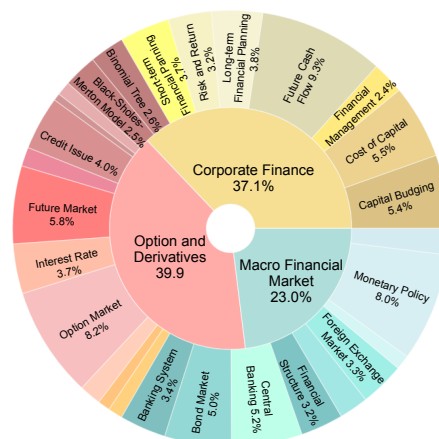

Figure 3: Finance concept distribution of XFIN-BENCH. Concepts with $\leq 2.5\%$ are not displayed.

(Meta, 2024a), Llama-3 models (Meta, 2024c), and Mixtral-7×8B (Jiang et al., 2024). All MLLMs allow text-only input except for Llama-3.2-Vision models, which we feed with a blank image in text-only tasks. Additionally, we evaluate the above models on the validation set of KnowledgeFMATH (Zhao et al., 2024) (200 examples), and a random sample of the test set of BizBench (Koncel-Kedziorski et al., 2024) (500 examples), for more comprehensive analysis and broader coverage of financial tasks. More details can be found in D.1.

We apply Chain-of-Thought (CoT) reasoning method (Wei et al., 2022) and use Accuracy as evaluation metrics in *statement judging* and *multi-choice question answering* tasks. In *financial calculation* task, we apply Program-of-Thought (PoT) method (Chen et al., 2023) in addition and use two evaluation metrics: 1) Accuracy of exact matching with correct answers, *i.e.* $Acc_{EM}$, and 2) Accuracy allowing the model's answer to be within 0.5% of the correct answer, *i.e.* $Acc_{ERR}$.

We further conduct knowledge augmentation analysis that allows access to external knowledge base during evaluation. We investigate 1) BM25 as sparse retriever and 2) OpenAI Ada embedding (OpenAI, 2022) as dense retriever to retrieve the top-$n$ question-relevant finance terms from knowledge bank, where $n$ is set to be 3. Further details of experiment setup can be found in §D.

## 3.2 MAIN RESULTS

We compare the performance of 18 leading models, including MLLMs and text-only LLMs, on XFINBENCH, BizBench and KnowledgeFMATH in Table 3. We also establish a human performance baseline with three graduate-level human experts in Finance over a random sample from test set with 1,000 examples. Further details can be found in §D.3.

Among MLLMs, claude-3.5-sonnet achieves the best performance with 64.1% accuracy on XFIN-BENCH, followed by gpt-4o with 63.6% accuracy who achieve the highest accuracy in visual-context questions, *i.e.*, 65.3%. On the text-only LLM side, o1 achieves the highest accuracy in almost all tasks of XFINBENCH, with 67.3% overall accuracy; however, it still falls 12.5% short of human performance, highlighting that there is a significant scope for further improvements on our benchmark. Open-source models with large parameter size, *i.e*, Llama-3.1-405B, achieves comparable performance with o1-mini and even outperforms gpt-4o-mini in text-only tasks on XFINBENCH. However, most open-source models achieve underwhelming performance, attributed to their lack of domain knowledge and mathematical reasoning ability. Additionally, enhanced performance on the *quantity extraction* task of BizBench and the *financial calculation* task of KnowledgeFMATH highlights XFINBENCH as a more sophisticated and challenging benchmark within the finance domain. $Acc_{ERR}$ scores in BizBench and KnowledgeFMATH are significantly higher than those in XFIN-BENCH for most models in calculating tasks. The model rankings across the three benchmarks are largely consistent, as indicated by the distribution of red cells in Table 3.

Table 3: Performance of models on XFINBENCH, BizBench and KnowledgeFMATH. Input: Q: question, I: image, [T]: tabular (optional). "Stmt judging" refers to *statement judging*; "MC question" refers to *multi-choice question*; "KFMATH" refers to KnowledgeFMATH. Evaluation metric for calculation is $Acc_{ERR}$. In each model setup, dark and light red cells have the highest and second highest scores in their column, respectively.

| Dataset | XFINBENCH | | | | | | BizBench | | | KFMATH | |
| --- | --- | --- | --- | --- | --- | --- | --- | --- | --- | --- | --- |
| Task | *Statement judging* | *Multi-choice question* | *Financial calculation* | | | *All* | *Multi-choice question* | *Quantity extraction* | | *Financial calculation* | |
| Reasoning | CoT | CoT | CoT | CoT | PoT | CoT | CoT | CoT | PoT | CoT | PoT |
| Input | Q | Q | Q,I | Q,[T] | Q,[T] | Q,[T] | Q | Q,[T] | Q,[T] | Q,[T] | Q,[T] |
| *Multimodal Large Language Models* | | | | | | | | | | | |
| gpt-4o | 84.0 | 91.5 | 65.3 | 49.6 | 45.9 | 63.6 | 80.1 | 64.3 | 69.6 | 58.5 | 51.0 |
| gpt-4o-mini | 76.5 | 86.8 | 54.8 | 40.5 | 40.3 | 57.4 | 69.5 | 71.3 | 73.5 | 47.0 | 46.0 |
| claude-3.5-sonnet | 84.3 | 94.2 | 63.7 | 49.6 | 49.0 | 64.1 | 83.0 | 64.9 | 63.0 | 59.0 | 55.0 |
| claude-3-opus | 79.0 | 91.2 | 50.7 | 42.9 | 41.2 | 59.7 | 77.3 | 47.9 | 33.4 | 51.0 | 46.5 |
| claude-3-haiku | 70.0 | 82.9 | 43.6 | 24.9 | 31.3 | 50.1 | 61.7 | 37.6 | 51.5 | 21.5 | 31.5 |
| gemini-1.5-pro | 76.3 | 86.5 | 50.8 | 38.8 | 42.8 | 57.3 | 75.2 | 66.3 | 30.6 / 61.3 | 54.5 | 58.5 |
| gemini-1.5-flash | 74.0 | 82.5 | 49.2 | 32.7 | 39.9 | 54.5 | 61.7 | 57.1 | 68.2 | 30.5 | 41.5 |
| Llama-3.2-90B-Vision | 57.4 | 70.9 | 47.6 | 20.0 | 18.8 | 42.0 | 68.1 | 39.6 | 24.2 | 24.0 | 28.5 |
| Llama-3.2-11B-Vision | 51.8 | 70.3 | 42.0 | 12.4 | 18.1 | 36.9 | 51.1 | 35.7 | 29.2 | 18.0 | 21.0 |
| *Text-only Large Language Models* | | | | | | | | | | | |
| o1 | 87.6 | 94.0 | | 63.0 | 51.3 | 67.3 | 89.4 | 62.1 | 60.7 | 68.5 | 50.0 |
| o1-mini | 81.0 | 90.0 | | 53.9 | 49.8 | 62.0 | 77.3 | 53.2 | 59.9 | 53.5 | 55.5 |
| deepseek-chat | 74.4 | 88.2 | | 46.9 | 47.9 | 59.6 | 72.3 | 71.6 | 56.5 | 53.0 | 51.0 |
| Llama-3.1-405B | 83.6 | 91.9 | | 41.5 | 31.7 | 61.9 | 78.0 | 59.9 | 47.9 | 46.5 | 27.5 |
| Llama-3.1-70B | 80.5 | 90.0 | | 37.2 | 26.9 | 59.3 | 78.7 | 67.4 | 45.4 | 44.0 | 30.0 |
| Llama-3-70B | 78.2 | 85.9 | | 30.2 | 21.1 | 56.1 | 70.2 | 60.2 | 15.6 | 33.0 | 24.0 |
| Llama-3.1-8B | 65.3 | 77.8 | | 18.5 | 20.3 | 45.5 | 56.7 | 56.3 | 47.1 | 20.0 | 25.0 |
| Llama-3-8B | 63.0 | 75.9 | | 14.0 | 14.9 | 42.9 | 55.3 | 44.6 | 34.3 | 14.0 | 14.5 |
| Mixtral-$8 \times 7$B | 26.1 | 29.9 | | 2.3 | 1.4 | 16.6 | 56.7 | 9.5 | 1.4 | 5.5 | 9.5 |
| *Human* | | | | | | | | | | | |
| Human | 90.9 | 92.1 | 81.1 | 65.6 / 78.6 | | 79.8 | 88.6 | 86.3 / 91.9 | | 73.5 / 85.0 | |

We observe that the PoT prompting method deteriorates the performance of most models in *financial calculation* task. To better analyze the reasons for these differing performance outcomes, we examine the execution rate of models under PoT prompting on XFINBENCH, measuring how many of the generated Python programs are executable (Zhao et al., 2024). Figure 5(b) illustrates the relationship between execution rate and accuracy $Acc_{ERR}$ across different models, indicating that the degraded performance when applying PoT prompting is attributable to the low execution rate. For instance, while Llama-3.1-405B achieves competitive performance using CoT prompting, it struggles to consistently generate executable Python solutions, leading to lower accuracy with PoT prompting. Interestingly, while o1's execution rate lags behind most close-source models, it achieves the highest accuracy score on $Acc_{ERR}$, witnessing its strong and efficient reasoning ability over complex tasks. We further report fine-grained results during evaluation in §E.

## 3.3 KNOWLEDGE AUGMENTATION METHOD

We evaluate the performance of models augmented with external knowledge base, and apply two types of retrievers to acquire the relevant knowledge term to the question, *i.e. BM25* and *Ada Embed*. Recalling that we have annotated the most relevant finance terms for each question, we design a *Oracle* setting, where models are provided with the *ground-truth* finance term(s) of each question.

We report the accuracy improvements of four models when augmented with a knowledge bank in Figure 4. For various retrieving settings, we find that the *Oracle* setting leads to the most robust improvements on most models, highlighting the high quality of our annotated dataset. Models employing a dense retriever based on Ada embedding achieve higher accuracy improvements compared to those using a sparse retriever with BM25, for most models. Furthermore, we report the accuracy improvements across five financial capabilities under *Oracle* setting in Figure 4(b). While the im-

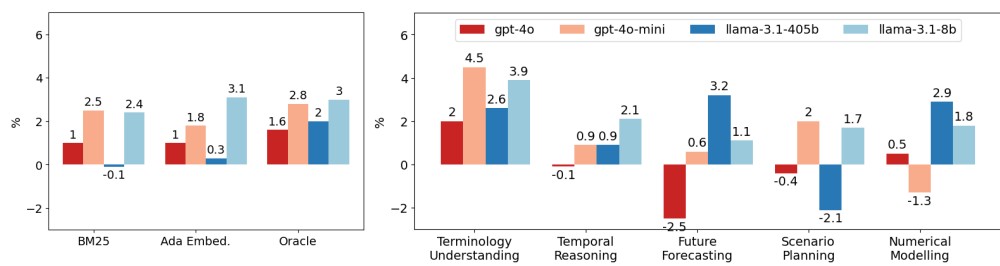

(a) Accuracy improvement across retrieving settings    (b) Accuracy improvement across five capability in Oracle setting

Figure 4: Accuracy improvements when augmented with external knowledge base. (a) displays the overall accuracy changes in different retrieving settings. (b) illustrates the accuracy changes across five capabilities for complex finance problem solving in *Oracle* setting.

provements in *terminology understanding* are consistently positive across all models, ground-truth knowledge augmentation does not always lead to better performance in advanced capabilities. The improvements across four advanced capabilities are inconsistent for most models, even experiencing negative changes, while the smallest open-source model, *i.e*, Llama-3.1-8B, have all positive improvements across all capabilities. Details of knowledge augmentation can be found in §E.

## 3.4 ERROR ANALYSIS

We conduct error analysis on two tasks, *i.e.* the *financial calculation* task and the visual-context *multiple-choice question answering* task, and knowledge augmentation method. For analysis on two tasks, we randomly select 400 and 100 samples from responses of best performers in each task, *i.e.*, o1 and gpt-4o. For analysis on knowledge augmentation method, we randomly select 100 samples from responses of gpt-4o that deliver wrong final answers under *Oracle* setting. Human annotators are then instructed to label various error types among these responses. Details can be found in §F.1.

**Error Analysis of Financial Calculation.** Based on our observation, two primary reasons of incorrect responses in calculating task are: 1) Rounding Error that exists in the intermediate calculating steps, and 2) Knowledge Misuse if applying wrong or incomplete finance formulas for calculation. Annotators are instructed to decide whether the reasoning path is correct and whether any error type exists in o1's responses. As illustrated in Figure 5(a), 55.2% of o1's response had correct reasoning path without intermediate rounding error or knowledge misuse - but might contain rounding error in the final step. Knowledge misuse appears more frequently in incorrect-reasoning responses, while rounding error often exists in correct reasoning process. For better illustration, we display an example of o1's response containing both two errors in Figure 6(b). In this example, o1 fails to use the primary property of American options, *i.e.* exercising the option before expiration date for profit maximization, and hence leads to unnecessary calculation in the following nodes. It also presents a rounding error when building binomial tree, which inevitably leads to an incorrect answer in the end. Additionally, we present a case of how knowledge augmentation could help improve gpt-4o's complex reasoning ability in finance task in Figure 6(a). The gold formula prompts to incorporate temporal and statistical reasoning abilities for calculation of future value.

**Error Analysis of Visual Context.** The error types identified in the visual-context *multiple-choice question answering* task are as follows: 1) Blindness (Rahmanzadehgervi et al., 2024), where the model struggles with identifying the position and/or intersection of two curves, and 2) Knowledge Misuse, occurring when irrelevant knowledge is introduced, thereby disrupting the reasoning path. Annotators are first instructed to determine if the explanation in the model's response is correct, partially correct, or wrong (Lu et al., 2024), considering both image description and reasoning process. For partially correct and wrong explanations, annotators are then asked to identify the two error types among gpt-4o's responses. As illustrated in Figure 7(a), 11.5% of its response had incorrect answer with incorrect explanations. Interestingly, we observe that gpt-4o responds with partial (9.6%) or completely (11.5%) incorrect explanations despite giving the correct answer. In Figure 7(b), we find that the model responds with correct reasoning but either blindness (49%) or knowledge misuse (6.1%), leaving an overall impression of being a wrong explanation. It is worth noting that 71.4% of its responses contain blindness in correct reasoning path, highlighting that blindness

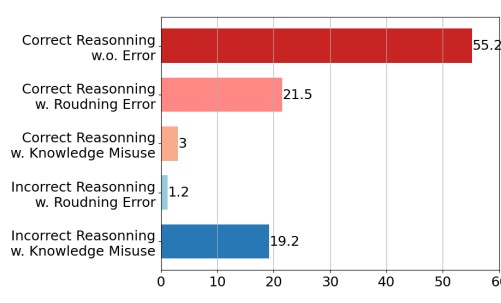
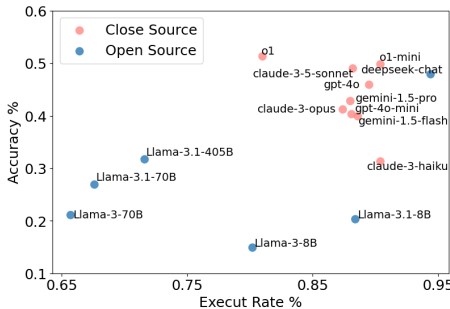

(a) Errors in reasoning process.

(b) Relation between accuracy score $Acc_{ERR}$ and execution rate in PoT.

Figure 5: Error analysis in *financial calculation*: (a) presents o1's errors in reasoning process; (b) shows relationship between accuracy score $Acc_{ERR}$ and executing rate under PoT setting.

---

**Question:** A stock price is currently $40. Assume that its expected return is 15% and its volatility is 25%. What is the expected value of the return over a two-year period with continuous compounding? (Unit: percent)

**Response from gpt-4o w.o. knowledge augmentation:**

Expected return 15% remains constant over the entire period. Unaware of temporal relation with future value Therefore, my answer is **15%**. in the statistical way

**Response from gpt-4o augmented with gold formula:**

Formula for continuously compounded return is: $x \sim \phi\left(\mu - \dfrac{\sigma^2}{2}, \dfrac{\sigma^2}{T}\right)$

$S_0 = 40, \mu = 0.15, \sigma = 0.25, T = 2$

$\mu - \dfrac{\sigma^2}{2} = 0.15 - \dfrac{0.25^2}{2} \approx 12\%$   Therefore, my answer is **12%**

**Correct Output:** 12
**Capability:** Temporal Reasoning; Statistical Reasoning

(a) Knowledge misuse before knowledge augmentation

---

**Question:** A three-month American call option on a stock has a strike price of $20. ... A dividend of $2 is expected in 1.5 months. What is the option price using a three-step binomial tree? (Unit: $)

**Response from o1:**

Ingoring Early Exercise at node time 2

$p_{up} = \dfrac{e^{r\Delta t} - d}{u - d} = 0.4982$

$p_{up} = 0.4993$   Rounding error in tree building

**Correct Output:** 0.674
**Capability:** Scenario Planning; Arithmetic Reasoning

(b) Incorrect reasoning w. Two error types

---

Figure 6: Two examples of calculation. In (a), gpt-4o misuses knowledge about stock's future price, and corrects itself when augmented with the gold formula; in (b) o1 fails to use early-exercise property of American option at node 2, and presents a rounding error when building binomial tree.

is a major source of errors in the generative foundation models (Rahmanzadehgervi et al., 2024). We present two examples of gpt-4o's responses to illustrate the two error types. In Figure 9(a), gpt-4o correctly identifies the temporal trend in the image, interpret its economic implication, and then analyze its effect on goods price. Correct image description and reasoning path in finance domain leads to the correct final answer. By contrast, in Figure 9(b), while gpt-4o outputs the correct final answer, its response contain both two error types, *i.e.*, misunderstanding of supply in bond market and blindness to the intersection of $R^{d2}$ and $R^s$ curves. Overall, our analysis of gpt-4o highlights its modes of failure, which could guide future foundation model design to address these issues.

**Error Analysis of Knowledge Augmentation.** We identify three error types when models are augmented with *ground-truth* finance term(s) but still fail to deliver the correct final answers: 1) Reasoning Error that appears in the model's reasoning process and has no direct relation to the augmented knowledge; 2) Over Thinking, in which case augmented knowledge provides direct solutions but the model reasons further steps that go out of the question's scope; 3) Over Reliance, in which case the model's reasoning process is entirely guided by augmented knowledge, foregoing simpler approaches to answering the question. As illustrated in Figure 8, most of wrong final answers for calculating questions, especially those requiring *temporal reasoning* and *numerical modelling* ca-

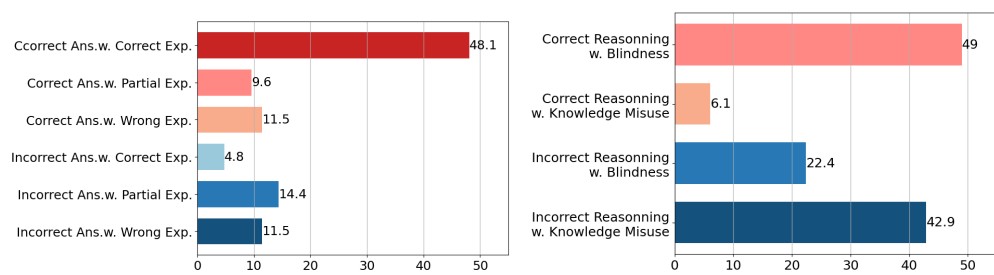

(a) Errors in answers and explanations.

(b) Types of wrong explanations.

Figure 7: Error analysis of GPT-4o in *multi-choice question answering* task with visual context: (a) presents errors in answers and explanations; (b) displays the details of wrong explanations. Notations: "Answer" is "Ans.", "Explanation" is "Exp.", and "Partially Correct" is "Partial".

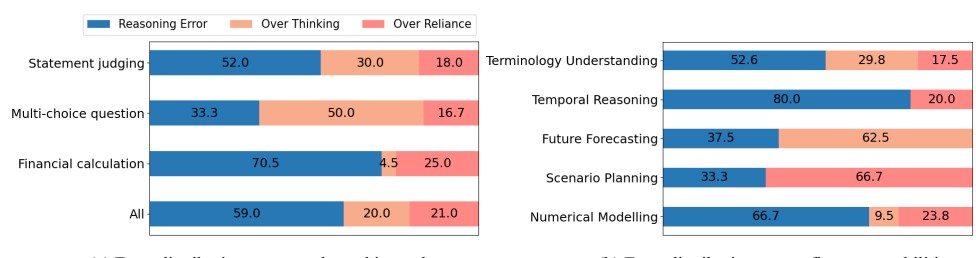

(a) Error distribution across tasks and in total

(b) Error distribution across finance capabilities

Figure 8: Error analysis of GPT-4o in *knowledge augmentation*: (a) presents proportions of each error type across three tasks and in total; (b) presents proportions across five finance capabilities.

pabilities, are caused by reasoning error that has little to do with augmented knowledge, such as rounding error. Over thinking is most frequently observed in multiple-choice questions requiring *future forecasting* capability, suggesting that GPT-4o exhibits a tendency to engage in deeper reasoning when addressing questions involving predictions of future events. Moreover, over reliance is most commonly encountered in questions requiring *scenario planning* capability, which emphasizes the model's ability to plan rather than strictly adhering to the instructions provided in the augmented knowledge. Detailed guidelines and cases studies of error analysis can be found in F.

Among the error types discussed above, blindness imposes greater demands on the visual-textual alignment capabilities of models. This limitation is likely attributable to the late-fusion approach (Alayrac et al., 2022; Liu et al., 2023) used for integrating vision into LLMs, suggesting that an early-fusion strategy (Team, 2024a; Tong et al., 2024) may offer a more effective solution. Errors such as rounding errors, knowledge misuse, and knowledge-augmentation errors could potentially be alleviated through more advanced prompting techniques, such as self-consistency CoT (Wang et al., 2023), least-to-most CoT (Zhou et al., 2023a), etc.

## 4 CONCLUSION

In this work, we introduced XFINBENCH, a novel benchmark designed to evaluate LLM's ability in solving complex, knowledge-intensive financial problems across diverse graduate-level topics with multi-modal context. We identified five core capabilities of LLMs using XFINBENCH, *i.e*, *terminology understanding*, *temporal reasoning*, *future forecasting*, *scenario planning*, and *numerical modelling*. Upon XFINBENCH, we conducted extensive experiments on 18 leading models. The result shows that o1 is the best-performing text-only model with an overall accuracy of 67.3%, but still lags significantly behind human experts with 12.5%. We further constructed a knowledge bank with 3,032 finance terms for knowledge augmentation method and conduct detailed error analysis across different tasks and models.

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

APPENDIX

## A  DATA COLLECTION GUIDELINES

### A.1  FINANCIAL AND MATHEMATICAL CAPABILITY DEFINITION

We define five core capabilities required for tackling complex finance problems in Table 4, along with their proportions. We also introduce five mathematical reasoning types from MATHEVISTA (Lu et al., 2024): *arithmetic reasoning*, *statistical reasoning*, *algebraic reasoning*, *numeric commonsense*, and *logical reasoning*. For annotation of questions in our *financial calculation* task with these mathematical reasoning types, we use a zero-shot prompt with definitions of these reasoning types, and leverage gpt-4o-mini to label each question with 1-to-2 mathematical reasoning type(s). See Table 5 for annotation results. The prompt template for annotation can be found in Appendix G.2.4.

Table 4: Definitions of five capabilities of solving complex, knowledge-intensive finance problem.

| Capability | Description |
| --- | --- |
| Terminology Understanding (56.1%) | It refers to the model's ability to accurately understand finance concepts, including standard financial terms, acronyms, accounting principles, various financial instruments, regulatory terminologies, and economic indicators. |
| Temporal Reasoning (21.7%) | It focuses on understanding temporal relations in time-based data, and making time-sensitive decisions. It often involves data over various time periods, like quarterly earnings reports, historical stock performance and future cash flow projections. |
| Future Forecasting (5.0%) | It involves predicting future values or trends of financial indicators such as output level, price level and inflation rates. It requires the model to use economic theories and quantitative methods to generate forecasts for strategic decision-making. |
| Scenario Planning (7.6%) | It is the process of generating and analyzing different possible future scenarios to assess their impact on financial decisions and strategies. It requires considering various uncertainties and variables to prepare for various outcomes. |
| Numerical Modelling (17.2%) | It involves creating structured representations of a company or product's financial performance. Related questions typically include financial statements like income statements, balance sheets, and cash flow statements. |

Table 5: Definitions of five mathematical reasoning types in Lu et al. (2024).

| Capability | Description |
| --- | --- |
| Arithmetic Reasoning (80.8%) | It covers the fundamental operations such as addition, subtraction, multiplication, division, and understanding of number properties. It may also include the ability to interpret numerical data in different forms. |
| Statistical Reasoning (77.9%) | It focuses on data interpretation and analysis, including measures (mean, median, mode), dispersion metrics (standard deviation, range), probability concepts, regression, correlation, and data inferences. It also identifies trends, outliers, and patterns. |
| Algebraic Reasoning (5.3%) | It encompasses understanding variables, equations, and the manipulation of expressions with polynomials and exponents. It also covers solving simple to complex equations, and grasping functions, their properties, and graphical depictions. |
| Numeric Commonsense (10.8%) | It involves intuitive understanding of daily numerical concepts, including understanding time differences, numerical judgment, and estimates. It covers temporal reasoning, spatial numeric assessments, and practical uses like budgeting and time reading. |
| Logical Reasoning (23,6%) | It focuses on critical thinking and deduction from provided information, including pattern recognition, sequence understanding, predictions, and statement evaluation. Key components include premises, conclusions, and the use of abstract reasoning. |

### A.2  FINANCIAL AND MATHEMATICAL CAPABILITY ANNOTATION

For five financial capabilities, we ask three human annotators to label each question in our dataset with 1-to-2 capability. A question will be labelled with one capability if at least two annotators choose this capability to label it. Specifically, questions that focus on the comprehension of financial terms and mathematical formulas are labeled as requiring *terminology understanding*. Questions necessitating the model's reasoning over time-series data, concepts, and mathematical formulas are categorized under *temporal reasoning*. When a question centers on predicting future trends, it is

marked as requiring *future forecasting*. For questions that involve analyzing potential future scenarios to aid in decision-making, the label *scenario planning* is used. Lastly, questions that involve creating structured representations of a company's financial performance using financial statements and informed assumptions are identified as needing *model building*.

For mathematical capabilities summarized in Lu et al. (2024), we leverage gpt-4o-mini for annotation. Specifically, we use the prompt template in Appendix G.2.4 to annotate each calculation problem in our dataset.

## A.3 EXAMPLES OF FINANCIAL CAPABILITY

Examples to display five capabilities for complex finance problem solving are shown in A.3.1, A.3.2, A.3.3, A.3.4, and A.3.5.

### A.3.1 EXAMPLES OF TERMINOLOGY UNDERSTANDING

---

**Example 1 of Terminology Understanding in *Statement Judging* task**

An investor holds a strip and believes that there will be a big jump in a stock price. He will earn a bigger profit when there is a large upward stock price move than a downward move.
Answer: False

---

**Example 2 of Terminology Understanding in *Multi-choice Question* task**

A bank is managing floating-rate deposits and fixed-rate loans, leading to asset-liability mismatch. Which one of the following swaps can help the bank offset risk?
A. Pay fixed and receive floating
B. Pay floating and receive fixed
C. Pay variable and receive fixed
Answer: A

---

### A.3.2 EXAMPLES OF TEMPORAL REASONING

---

**Example 1 of Temporal Reasoning in *Financial Calculation* task**

You own 1,000 shares of stock in Avondale Corporation. You will receive a $1.50 per share dividend in one year. In two years, Avondale will pay a liquidating dividend of $45 per share. The required return on Avondale stock is 15 percent. What would be the equal dividend per share in each of the next two years to have the same present value as the current share price?
(Unit: dollar)
Answer: 21.73

---

**Example 2 of Temporal Reasoning in *Financial Calculation* task**

The price of a European call that expires in six months and has a strike price of $30 is $2. The underlying stock price is $29, and a dividend of $0.50 is expected in two months and again in five months. Interest rates (all maturities) are 10%. If the stock price is above $30 in six months, what is the present value of the profit? (Unit: dollar)
Answer: 0.49

---

### A.3.3 EXAMPLES OF FUTURE FORECASTING

> **Example 1 of Future Forecasting in *Multi-choice Question* task**
>
> Both Keynes' and Friedman's theories of the demand for money discuss the impact of interest rates on money demand. According to Keynes model, which one of the following outcomes happens when interest rates rise?
> A. Demand for money decreases
> B. Demand for money increases
> C. Demand for money stays unchanged
> Answer: A

> **Example 2 of Future Forecasting in *Multi-choice Question* task**
>
> Interest rates tend to change in response to the increase or decrease of aggregate output during economic booms and recessions. Which one of the following actions might banks take when output rises during a boom?
> A. Freeze the level of their excess reserves
> B. Reduce the level of their excess reserves
> C. Increase the level of their excess reserves
> Answer: B

### A.3.4 EXAMPLES OF SCENARIO PLANNING

> **Example 1 of Scenario Planning in *Multi-choice Question* task**
>
> A trader sells a strangle by selling a call option with a strike price of $50 for $3 and selling a put option with a strike price of $40 for $4. Within which one of the following price ranges of the underlying asset does the trader make a profit?
> A. Between $33 and $57
> B. Between $30 and $50
> C. Between $40 and $60
> Answer: A

> **Example 2 of Scenario Planning in *Financial Calculation* task**
>
> On May 8, 2013, an investor owns 100 Google shares. The share price is about $871 and a December put option with a strike price of $820 costs $37.50. The investor is comparing two alternatives to limit downside risk. The first involves buying one December put option contract with a strike price of $820. The second involves instructing a broker to sell the 100 shares as soon as Google's price reaches $820. How much will the investor pay to buy one December put option contract with a strike price of $820? (Unit: dollar)
> Answer: 3750

### A.3.5 EXAMPLES OF NUMERICAL MODELLING

> **Example 1 of Numerical Modelling in *Financial Calculation* task**
>
> Bedrock Gravel Corp.'s 2007 income statement shows the following information: sales = $162,000; costs = $93,000; other expenses = $5,100; depreciation expense = $8,400; interest expense = $16,500; taxes = $14,820; dividends = $9,400. Additionally, the firm issued $7,350 in new equity during 2007 and redeemed $6,400 in outstanding long-term debt. What is the 2007 operating cash flow? (Unit: dollar)
> Answer: 49080

> **Example 2 of Numerical Modelling in *Financial Calculation* task**
>
> Winnebagel Corp. currently sells 30,000 motor homes per year at $45,000 each, and 12,000 luxury motor coaches per year at $85,000 each. The company wants to introduce a new portable camper to fill out its product line; it hopes to sell 19,000 of these campers per year at $12,000 each. An independent consultant has determined that if Winnebagel introduces the new campers, it should boost the sales of its existing motor homes by 4,500 units per year, and reduce the sales of its motor coaches by 900 units per year. What is the annual sales figure due solely to the new portable camper product line? (Unit: dollar)
> Answer: 228000000

## A.4 RELATED WORKS OF VISUAL-CONTEXT QUESTIONS

Visual-context examples in XFINBENCH are closely related to multi-modal benchmarks that involve chart understanding and reasoning, as shown in Table 6. Most related multi-modal benchmarks focus on descriptive question of charts that evaluates model's perception ability. For example, multi-discipline multi-modal benchmarks, *i.e,*, MMMU (Yue et al., 2024), MMLU-Pro (Wang et al., 2024a) and MathVista (Lu et al., 2024), create descriptive questions around the visual information of charts, such as locating the number of a bar and finding the trend of a line. Although they emphasize domain-specific knowledge for tackling their tasks, they stop at reading the technical terms that appear in the charts in finance domain. Moreover, there are many chart benchmarks that heavily rely on datasets from finance domain, including ChartQA (Masry et al., 2022), MMC (Liu et al., 2024) and CharXiv (Wang et al., 2024b). They focus on both chart understanding and reasoning, while their reasoning tasks focus on multiple-step reasoning over the visual information, instead of domain-specific reasoning.

Visual-context questions in our XFINBENCH, however, require not only reasoning over visual information of chart, but also interpreting the financial implications of data presented in the chart. For example, the chart in Figure 9 (a) evaluates model's ability to find out the trend of exchange rate over time and then link it with the effect of exchange rate on the good price across two countries. The former step focus on reasoning over visual information like previous works do, while the latter one requiring complex financial reasoning. Despite the small size of our visual-context questions, our work is the first to explore the model's potential of applying finance-domain knowledge to complex chart reasoning. Examples of our visual-context questions are displayed in A.5.

Table 6: Comparison of visual-context questions in XFINBENCH with existing multi-modal datasets. "# Image" refers to number of image; "# Ques." refers to number of question; "NA" indicates not reported in the main body of the paper. For tasks, "MCQ" and "OQ" stand for multiple-choice question and open question, respectively.

| Dataset | # Image / # Ques. | Domain | Image Type | Task | Finance Component | | Source |
|---|---|---|---|---|---|---|---|
| | | | | | Descriptive-Question | Financial-Reasoning | |
| MMMU | 11,550 / 11,264 | Art, Finance, Science, Medicine, Social Science, Tecnology | Diagram, Table, Chart, Geometric, Science Photo | MCQ, OQ | ✓ | ✗ | Textbooks, Internet, w. CrowdSource |
| MMLU-Pro | NA / 12,032 | Finance, Science, Medicine, Technology | Diagram, Table, Chart, Geometric, Science Photo | MCQ | ✓ | ✗ | Existing Datasets, w. CrowdSource |
| MathVista | 5,487 / 6,141 | Finance, Science, Medicine, Technology | Diagram, Table, Chart, Geometric, Science Photo, Natural Image | MCQ, OQ | ✓ | ✗ | Existing Datasets, w. CrowdSource |
| ChartQA | 21,945 / 32,719 | Finance, Social Science | Chart | OQ | ✓ | ✗ | Internet, w. CrowdSource, w. Machine (T5) |
| MMC | 2,126 / 1,063 | Finance, Science | Chart | MCQ, OQ | ✓ | ✗ | Existing Datasets, Internet |
| CharXiv | 2,323 / 11,615 | Finance, Science, Technology | Chart | MCQ | ✓ | ✗ | Internet, w. CrowdSource |
| Visual-context XFINBENCH | 64 / 146 | Finance | Chart | MCQ | ✓ | ✓ | Textbook, w. CrowdSource, w. Machine (GPT-4o) |

## A.5 Examples of Visual-context Questions

---

**Example 1 of Visual-context question in *Multi-choice Question* task that evaluates Scenario Planning capability**

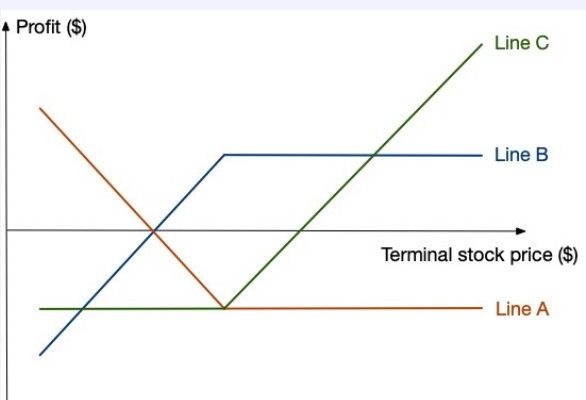

Suppose that a June put option to sell a share for $60 costs $4 and is held until June. Which line in the attached figure best describes the relationship between the option's profit and the stock price?
A. Line A
B. Line B
C. Line C
Answer: B

---

**Example 2 of Visual-context question in *Multi-choice Question* task that evaluates Future Forecasting capability**

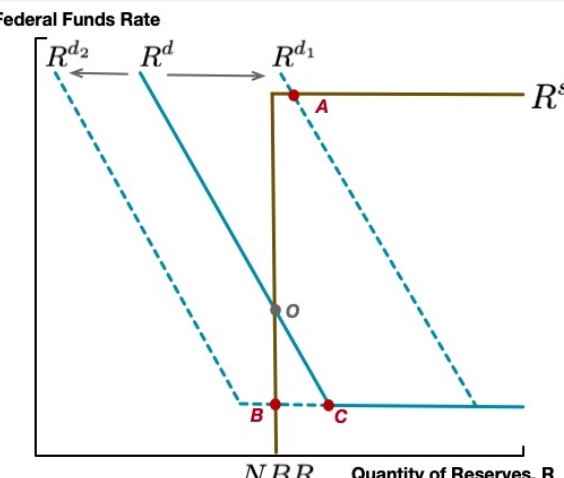

Suppose the economy is surprisingly strong, leading to an increase in the amount of checkable deposits. Given the supply and demand diagram of reserve market, which one of the following points will the original balance point O move to?
A. Point A
B. Point B
C. Point C
Answer: A

---

Example 3 of Visual-context question in *Financial Calculation* task that evaluates Temporal Reasoning capability

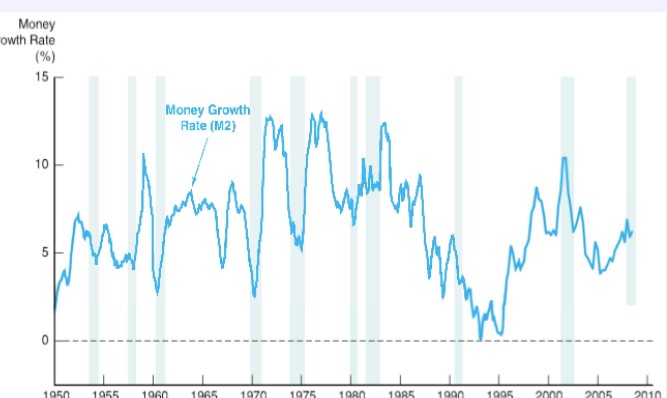

The shaded areas in the attached figure represent recessions. What is the relationship between the rate of money growth and recessions as indicated in this figure?
A. The rate of money growth has declined before every recession;
B. The rate of money growth has little correlation with the recession periods;
C. The rate of money growth has increased before every recession.
Answer: A

Example 1 of Visual-context question in *Financial Calculation* task that evaluates Numerical Modelling capability

|  | 2006 | 2007 |
|---|---|---|
| Sales | $ 4,822 | $ 5,390 |
| Depreciation | 692 | 723 |
| Cost of goods sold | 1,658 | 1,961 |
| Other expenses | 394 | 343 |
| Interest | 323 | 386 |
| Cash | 2,528 | 2,694 |
| Accounts receivable | 3,347 | 3,928 |
| Short-term notes payable | 488 | 478 |
| Long-term debt | 8,467 | 10,290 |
| Net fixed assets | 21,203 | 22,614 |
| Accounts payable | 2,656 | 2,683 |
| Inventory | 5,951 | 6,370 |
| Dividends | 588 | 674 |

For 2007, calculate the cash flow from assets, cash flow to creditors, and cash flow to stockholders based on financial data from the table. What is the value of total liability and equity of this firm during 2006? (Unit: dollar)
Answer: 33029

## A.6 EXAMPLES OF TERM DEFINITIONS

Example 1 of finance term and definition

Term: Two-stage growth model for common stock valuation
If the dividend grows at rate $g_1$ for $t$ periods and then grows at rate $g_2$ thereafter, then the price can be written as: $P_0 = \frac{D_1}{R-g_1} \times \left[1 - \left(\frac{1+g_1}{1+R}\right)^t\right] + \frac{P_t}{(1+R)^t}$, where $P_t = \frac{D_{t+1}}{R-g_2} = \frac{D_0 \times (1+g_1)^t \times (1+g_2)}{R-g_2}$, $D_1$ is the next dividend, and $R$ is the required return.

**Example 2 of finance term and definition**

Term: Total credit cost curve of optimal credit policy
The trade-off between granting credit and not granting credit isn't hard to identify, but it is difficult to quantify precisely. As a result, $\cdots$ The sum of the carrying costs and the opportunity costs of a particular credit policy is called the total **credit cost curve**. We have drawn such a curve. There is a point where the total credit cost is minimized. This point corresponds to the optimal amount of credit or, equivalently, the optimal investment in receivables.\n\n If the firm extends more credit than this minimum, the additional net cash flow from new customers will not cover the carrying costs of the investment in receivables. If the level of receivables is below this amount, then the firm is forgoing valuable profit opportunities.\n\n In general, the costs and benefits from extending credit will depend on characteristics of particular firms and industries. All other things being equal, for example, it is likely that firms with (1) excess capacity, (2) low variable operating costs, and (3) repeat customers will extend credit more liberally than other firms. See if you can explain why each of these characteristics contributes to a more liberal credit policy.

**Example 3 of finance term and definition**

Term: Open market operations for control of Monetary Base
The Federal Reserve exercises control over the monetary base through its purchases or sale of government securities in the open market, called **open market operations**, and through its extension of discount loans to banks. A purchase of bonds by the Fed is called an **open market purchase**, and a sale of bonds by the Fed is called an **open market sale**.

**Example 4 of finance term and definition**

Term: Exchange-rate targeting
Targeting the exchange rate is a monetary policy strategy with a long history. It can take the form of fixing the value of domestic currency to a commodity such as gold, the key feature of the gold standard described earlier in the chapter. More recently, fixed exchange rate regimes have involved fixing the value of the domestic currency to that of a large, low-inflation country like the United States (the anchor country). Another alternative is to adopt a crawing target or peg, in which a currency is allowed to depreciate at a steady rate so that the inflation rate in the pegging country can be higher than that of the anchor country.

**Example 5 of finance term and definition**

Term: American call option
Black suggests an approximate procedure for taking account of early exercise in call options. This involves calculating, as described earlier in this section, the prices of European options that mature at times $T$ and $t_n$, and then setting the American price equal to the greater of the two.15 This is an approximation because it in effect assumes the option holder has to decide at time zero whether the option will be exercised at time $T$ or $t_n$.

**Example 6 of finance term and definition**

Term: Interest rates in convexity adjustment

Consider first an instrument that provides a payoff dependent on a bond yield observed at the time of the payoff. Usually the forward value of a variable $S$ is calculated with reference to a forward contract that pays off $S_T - K$ at time $T$. It is the value of $K$ that causes the contract to have zero value. $\cdots$ The relationship between the price of this bond and its yield is $G(y) = \frac{1}{1+y\tau}$ From equation (3.1), $E_T(R_T) = R_0 - \frac{1}{2}R_0^2\sigma_R^2 T\frac{G''(R_0)}{G'(R_0)}$ or $E_T(R_T) = R_0 + \frac{R_0^2\sigma_R^2\tau T}{1+R_0\tau}$ (3.2) where $R_0$ is the forward rate applicable to the period between $T$ and $T^*$ and $\sigma_R$ is the volatility of the forward rate. The value of the instrument is therefore $P(0,T)L\tau\left[R_0 + \frac{R_0^2\sigma_R^2\tau T}{1+R_0\tau}\right]$.

## B    DETAILED DATA CONSTRUCTION

### B.1    SOURCE DATA

The details of textbooks are displayed in Table 7. During data collection, annotators are instructed to adhere to copyright and license regulations, avoiding data from sites prohibiting copy and redistribution.

Table 7: Details of textbooks as source data.

| Textbook | Authors | Version | # Chapters |
|---|---|---|---|
| Fundamentals of Corporate Finance | Stephen A. Ross | 8 | 22 |
| Options, Futures and Other Derivatives | John C. Hull | 9 | 32 |
| The Economics of Money Banking and Financial Markets | Frederic S. Mishkin | 9 | 25 |

### B.2    QA TASK AND AUTOMATIC ANNOTATION

We leverage GPT-4o to process after-class questions under a generate-then-verify framework (Zhang et al., 2024). Few-shot prompt templates for generate-then-verify framework are in §G.

For the generation stage, examples in the prompt template illustrate the rules of transforming open-ended questions into those with clear final answers. For *statement judging* task, rules of creating false statements are: 1) antonym substitution, such as small → big; 2) object position interchange, such as "A is red and B is blue" → "B is red and A is blue"; 3) adjective modification, such as "it is possible" → "it is impossible", etc. For *multi-choice question answering* task, we follow STARC (Berzak et al., 2020) rules to design two misleading choices that are mutually exclusive to but share the similar wording and length with the correct choice. For *financial calculation* task, calculation questions usually have a series of sub-questions that share the same solution in the gold answer but have different final answers. In this case, GPT-4o simply split the question into independent questions with clear final answers. Furthermore, to ensure that the generated question contain necessary information to get its final answer, we ask GPT-4o to extract the context in the after-class question first, and then extract the question and its final answer (see examples in prompt templates).

For the verification stage, rules for discarding unqualified questions are illustrated in the prompt templates in §G.

### B.3    KNOWLEDGE BANK CONSTRUCTION AND ANNOTATION

We collect finance terms from the subject index at the end of each textbook, and manually extract their definitions from the chapter's content. Specifically, for each term, we locate its corresponding pages indicated in the subject index, and collect the paragraphs related to this term. There are two common cases during this process: (1) the term's name is the title of a subsection, so its related paragraphs are the main content of this subsection; (2) the term's definition in the corresponding page is within a highlighted box, so we only collect the information within the box. Mathematical expressions and tabular information are also collected if any, while visual context of terms is not saved in our dataset. When retrieving relevant terms of a question, we concatenate the names of terms with their definitions for representing each term in the abstract space. It is worth noting that some terms may share the same pages, indicating that they share the same definition. Examples of term and definition are shown in A.6.

To bridge questions and finance terms, three annotators are asked to identify 1-to-3 relevant finance terms from the knowledge bank to each question in XFINBENCH. For each question, annotators search for the relevant terms from those in the same textbook and chapter with this question. If the term is included, they verify its context and details for relevance. A finance term would only be annotated to the question when at least two annotators agree on the high relevance. Finally, a question has 1.3 finance term on average.

## C  MORE DATASET ANALYSIS

### C.1  HUMAN QUALITY VALIDATION

We conduct a comprehensive validation protocol to ensure the high quality of the annotated data. For each annotated question, we assign our three evaluators to validate whether: 1) the question contains complete information in the original question to get the final answer; 2) the final answer is correct given the original answer; 3) the associated knowledge terms are helpful for answering the question. We ask the evaluators to rate all examples in the test and validation sets of XFINBENCH on a scale of 1 to 5 individually. During this process, human evaluators are accessible to the corresponding after-class questions with gold answers and the knowledge bank. The result is illustrated in Table 8, indicating the high quality of our dataset.

Table 8: Human evaluation over the test and validation sets of XFINBENCH. Three evaluators are asked to rate the examples on a scale of 1 to 5 individually. In each dimension, we report the proportions of examples with average scores in different ranges.

| Score | Question Fluency | Question Completeness | Answer Correctness | Knowledge Helpfulness |
|---|---|---|---|---|
| %S = 5 | 92.9 | 95.2 | 96.3 | 94.1 |
| %S ≥ 4 | 97.1 | 97.7 | 98.0 | 96.8 |
| %S ≥ 3 | 99.4 | 99.3 | 99.6 | 99.8 |
| %S ≥ 2 | 99.4 | 99.4 | 99.8 | 99.9 |
| %S ≥ 1 | 100.0 | 100.0 | 100.0 | 100.0 |

We then collect examples that have at least one dimension score less than 4 in the test set to further reveal the data quality of XFINBENCH. We get 209 examples eventually and illustrate their common problems in Table 9. We further report the performance of models after filtering out these examples in Table 10, and find that the changes are almost within 1% and have little effect on the overall ranking in Table 3.

Table 9: Common problems of examples with at least one dimension score less than 4 in the test set of XFINBENCH.

| Dimension | Common Problem | Examples |
|---|---|---|
| Question Fluency | (1) There is overlap in different parts of the question, causing it not easy to read; (2) There is too much information in the question that disturbs the model; and (3) The question style does not correspond to its task. | For (1) and (2), "Some investors have obligations that are denominated in dollars; i.e., they are nominal. Their primary concern is that an investment provides the needed nominal dollar amounts. Pension funds often do not plan for pension payments many years in the future". For (3), as a question in *financial calculation* task, "Red Zeppelin Corporation follows ... for the coming year are $760,000. Will Red Zeppelin pay a dividend if the planned investment outlays for the coming year are $760,000?" |
| Question Completeness | (1) The question cites previous information (e.g., examples, snapshots in the chapter's main content); (2) The question only mentions the abbreviation of professional term; and (3) The question does not assume that other variables are constant when discussing changes. | For (1), "In our capital budgeting examples, we assumed that a firm would recover all of the working capital it invested in a project. Current liabilities will not be paid". For (2), "Consider the relationship between bond price, coupon rate, YTM, and current yield. For premium bonds, the current yield exceeds the YTM". For (3), "Unexpected fluctuations in deposits impact the demand for reserves. Changes in banks' desire to hold excess reserves do not affect the demand curve for reserves". |
| Answer Correctness | The answer to the calculating question contains unit, like $ and %. | 13.4%; $ 51.1. |
| Knowledge Helpfulness | The terms do not cover all aspects of the question. | "BlueSky lengthened its payables period to 'control costs and optimize cash flow.' With this change, BlueSky will likely need more short-term borrowing from other sources, increasing its interest expense.". The *ground-truth* term of this question is Payables Period, which does not introduce the meaning of Short-term Borrowing. |

Table 10: Performance of four models on the test set XFINBENCH with and without examples that have at least one dimension score less than 4. "wLQ" refers to data with these low-quality examples, and "woLQ" refers to data without them.

| Task | Statement judging | | MC question | | Financial calculation | | All | |
|---|---|---|---|---|---|---|---|---|
| Data | wLQ | woLQ | wLQ | woLQ | wLQ | woLQ | wLQ | woLQ |
| gpt-4o-2024-05-13 | 84.0 | 84.3 | 91.5 | 91.5 | 31.8 / 49.6 | 30.8 / 47.7 | 63.6 | 63.7 |
| gpt-4o-mini-2024-07-18 | 76.5 | 76.4 | 86.8 | 86.9 | 26.5 / 40.5 | 26.1 / 39.9 | 57.4 | 57.6 |
| meta-llama-3.1-405b-instruct | 83.6 | 83.9 | 91.9 | 91.9 | 28.1 / 41.5 | 25.5 / 37.9 | 61.9 | 61.4 |
| Meta-Llama-3.1-8B-Instruct | 65.3 | 65.4 | 77.8 | 78.3 | 12.8 / 18.5 | 12.5 / 18.5 | 45.5 | 45.9 |

## C.2 DETAILED DATASET STATISTICS

The distribution of question over test and validation sets are shown in Table 11. The distribution of five capabilities for complex finance problem solving over three tasks are shown in Table 12.

Table 11: Distribution of task and capability in the test and validation set.

| Task | Test | Validation | Capability | Test | Validation |
|---|---|---|---|---|---|
| *Statement judging* | 1,360 | 436 | *Terminology understanding* | 1,814 | 582 |
| *Multi-choice question answering* | 592 | 169 | *Temporal reasoning* | 703 | 222 |
| *Financial calculation* | 1,283 | 396 | *Future forecasting* | 162 | 44 |
| | | | *Scenario planning* | 246 | 69 |
| | | | *Numerical modelling* | 557 | 188 |

Table 12: Distribution of questions in each finance capability (row) across three tasks (column).

| Capability | Statement judging | Multi-choice question answering | Financial calculation |
|---|---|---|---|
| *Terminology Understanding* | 74.7 | 24.3 | 1.0 |
| *Temporal Reasoning* | 3.9 | 6.6 | 89.5 |
| *Future Forecasting* | 22.8 | 45.6 | 31.6 |
| *Scenario Planning* | 3.2 | 8.3 | 88.6 |
| *Numerical Modelling* | 0.0 | 1.2 | 98.8 |

# D MORE EXPERIMENT SETUP

## D.1 EVALUATION ON BIZBENCH AND KNOWLEDGEFMATH

For BizBench (Koncel-Kedziorski et al., 2024), we randomly sample 500 examples from its test set. The reason why we select BizBench is that it covers most of previous finance dataset like TAT-QA and FinQA, and includes *quantity extraction* task that requires extracting numbers from contextual materials and conducting simple numerical reasoning. Additionally, we do not include SEC-NUM task of BizBench in our experiment due to its incomplete representation of questions.

For KnowledgeFMATH (Zhao et al., 2024), we use its validation set with 200 examples and ground truths released. The reason why we select KnowledgeFMATH is that it first introduces more complex numerical-reasoning questions than *quantity extraction* task in finance domain. While our XFINBENCH is more complex and challenging for both MLLM and text-only LLM, it is still worth evaluating our baselines on KnowledgeFMATH for more comprehensive study.

## D.2 MODEL HYPERPARAMTERS

The hyperparameters for the experiments are set to their default values unless specified otherwise. Table 13 detail specific generation parameters for the various large multimodal models (LMMs) and large language models (LLMs) we evaluated. Additionally, Open Ada embedding used in our experiment is `text-embedding-ada-002`.

Table 13: Generating parameters for vaious models.

| Model | Generation Setup |
|---|---|
| o1 | model=`o1-preview-2024-09-12`, max tokens=1024 |
| o1-mini | model=`o1-mini-2024-09-12`, max tokens=1024 |
| gpt-4o | model=`gpt-4o-2024-05-13`, max tokens=1024 |
| gpt-4o-mini | model=`gpt-4o-mini-2024-07-18`, max tokens=1024 |
| claude-3-5-sonnet | model=`claude-3-5-sonnet-20240620`, max tokens=1024 |
| claude-3-opus | model=`claude-3-opus-20240229`, max tokens=1024 |
| claude-3-haiku | model=`claude-3-haiku-20240307`, max tokens=1024 |
| gemini-1.5-flash | model=`gemini-1.5-flash`, max tokens=1024 |
| gemini-1.5-pro | model=`gemini-1.5-pro`, max tokens=1024 |
| deepseek-chat | model=`deepseek-chat`, max tokens=1024 |
| Llama-3.2-90B-Vision | model=`Meta-Llama-3.2-90B-Vision-Instruct`, max tokens=1024 |
| Llama-3.2-11B-Vision | model=`Meta-Llama-3.2-11B-Vision-Instruct`, max tokens=1024 |
| Llama-3.1-405B | model=`Meta-Llama-3.1-405B-Instruct`, max tokens=1024 |
| Llama-3.1-70B | model=`Meta-Llama-3.1-70B-Instruct`, max tokens=1024 |
| Llama-3.1-8B | model=`Meta-Llama-3.1-8B-Instruct`, max tokens=1024 |
| Llama-3-70B | model=`Meta-Llama-3-70B-Instruct`, max tokens=1024 |
| Llama-3-8B | model=`Meta-Llama-3-8B-Instruct`, max tokens=1024 |
| Mixtral-$8 \times$ 7B | model=`Mixtral-8x7B-Instruct-v0.1`, max tokens=1024 |

## D.3 HUMAN PERFORMANCE

We conducted a study to evaluate human performance in XFINBENCH. We randomly sampled 1,000 questions from test set of XFINBENCH, with 400 of *statement judging* task, 170 of *multi-choice question answering* task, and 430 of *financial calculation* task. Each question was then assigned to three human experts, all of whom have finance master degrees and have studied the courses covering three textbooks in our source data. None of them is involved in the dataset construction work. The human evaluation is conducted in a close-book setting, and allows standard calculators (not the financial ones). For each question in *statement judging* and *multi-choice question answering* tasks, they must complete each question within five minutes, while in *financial calculation*, the limit is ten minutes due to more reasoning process required in mathematical reasoning.

# E  MORE EXPERIMENT RESULTS

## E.1  RESULTS ACROSS DOMAIN CAPABILITY

We report the performance of models across five capability required by solving complex, knowledge-intensive finance problems in Table 14. Additionally, we report the performance of models across five mathematical reasoning types covered by *financial calculation* task in Table 15.

Table 14: Performance of models across five capabilities for complex finance problem solving.

| Model | Terminology Understanding | Temporal Reasoning | Future Forecasting | Scenario Planning | Numerical Modelling |
|---|---|---|---|---|---|
| gpt-4o-2024-05-13 | 85.4 | 22.6 | 62.3 | 32.9 | 38.8 |
| gpt-4o-mini-2024-07-18 | 78.4 | 18.9 | 58.0 | 28.9 | 33.0 |
| claude-3-5-sonnet-20240620 | 86.5 | 22.8 | 63.6 | 43.1 | 44.2 |
| claude-3-opus-20240229 | 81.5 | 19.3 | 53.1 | 37.0 | 41.3 |
| claude-3-haiku-20240307 | 72.4 | 12.8 | 40.1 | 25.6 | 26.6 |
| gemini-1.5-flash | 75.6 | 16.4 | 54.3 | 28.5 | 34.5 |
| gemini-1.5-pro | 78.7 | 20.2 | 53.7 | 34.6 | 36.8 |
| o1-preview-2024-09-12 | 88.9 | 24.8 | 74.7 | 45.0 | 45.8 |
| o1-mini-2024-09-12 | 83.0 | 21.4 | 66.3 | 38.7 | 41.8 |
| Meta-Llama-3.1-405B-instruct | 85.3 | 16.1 | 70.5 | 34.5 | 33.8 |
| Meta-Llama-3.1-70B-Instruct | 82.6 | 15.7 | 66.3 | 31.5 | 36.2 |
| Meta-Llama-3.1-8B-Instruct | 68.0 | 7.9 | 50.5 | 18.9 | 19.2 |
| deepseek-chat | 77.7 | 19.5 | 63.2 | 37.4 | 42.5 |
| Meta-Llama-3-70B-instruct | 79.9 | 11.2 | 61.1 | 30.3 | 33.3 |
| Human | 91.0 | 66.5 | 86.2 | 66.7 | 66.0 |

Table 15: Performance of models across five mathematical reasoning types (Lu et al., 2024).

| Model | Terminology Understanding | | Temporal Reasoning | | Future Forecasting | | Scenario Planning | | Numerical Modelling | |
|---|---|---|---|---|---|---|---|---|---|---|
| | Acc | Acc_err | Acc | Acc_err | Acc | Acc_err | Acc | Acc_err | Acc | Acc_err |
| gpt-4o-2024-05-13 | 29.2 | 32.1 | 23.4 | 26.8 | 23.5 | 23.5 | 29.5 | 33.1 | 33.7 | 39.9 |
| gpt-4o-mini-2024-07-18 | 24.2 | 26.6 | 19.8 | 22.1 | 17.6 | 17.6 | 24.5 | 27.3 | 29.4 | 34.0 |
| claude-3-5-sonnet-20240620 | 32.9 | 32.9 | 27.1 | 27.1 | 23.5 | 23.5 | 41.7 | 41.7 | 38.3 | 38.3 |
| claude-3-opus-20240229 | 28.6 | 28.6 | 23.4 | 23.4 | 27.9 | 27.9 | 35.3 | 35.3 | 35.6 | 35.6 |
| claude-3-haiku-20240307 | 18.0 | 18.0 | 14.1 | 14.1 | 16.2 | 16.2 | 17.3 | 17.3 | 21.8 | 21.8 |
| gemini-1.5-flash | 25.1 | 25.1 | 19.9 | 19.9 | 22.1 | 22.1 | 27.3 | 27.3 | 30.4 | 30.4 |
| gemini-1.5-pro | 28.4 | 28.4 | 21.9 | 21.9 | 25.0 | 25.0 | 32.4 | 32.4 | 32.7 | 32.7 |
| o1-preview-2024-09-12 | 38.5 | 38.5 | 31.7 | 31.7 | 25.0 | 25.0 | 40.3 | 40.3 | 40.9 | 40.9 |
| o1-mini-2024-09-12 | 33.5 | 33.5 | 27.7 | 27.7 | 25.0 | 25.0 | 36.0 | 36.0 | 37.3 | 37.3 |
| Meta-Llama-3.1-405B-instruct | 27.0 | 27.0 | 22.0 | 22.0 | 17.6 | 17.6 | 27.3 | 27.3 | 33.3 | 33.3 |
| Meta-Llama-3.1-70B-Instruct | 27.4 | 27.4 | 20.9 | 20.9 | 22.1 | 22.1 | 30.2 | 30.2 | 31.4 | 31.4 |
| Meta-Llama-3.1-8B-Instruct | 13.4 | 13.4 | 10.2 | 10.2 | 11.8 | 11.8 | 13.7 | 13.7 | 17.8 | 17.8 |
| deepseek-chat | 33.3 | 33.3 | 25.9 | 25.9 | 25.0 | 25.0 | 35.3 | 35.3 | 40.3 | 40.3 |
| Meta-Llama-3-70B-instruct | 23.0 | 23.0 | 17.1 | 17.1 | 20.6 | 20.6 | 23.7 | 23.7 | 30.4 | 30.4 |
| human | 67.5 | 78.7 | 64.0 | 76.1 | 52.9 | 82.4 | 65.1 | 81.4 | 65.1 | 81.6 |

## E.2  RESULTS ACROSS KNOWLEDGE AUGMENTATION METHODS

We report the performance of four models with different retrieving settings in Table 16. We design an evaluation metrics of retrievers, *i.e.*, the accuracy of retrievers locating at least 1 gold terms, annotated by human experts, from the knowledge bank. Dense retriever based on Ada embedding achieve higher accuracy than sparse retriever using BM25 over all tasks, and yield better performance of models under most circumstances. This finding illustrates that improving the question-relevance of

incorporated knowledge can consistently improve the LLMs' performance. Additionally, we report their performance across five financial capability in *Oracle* setting in Table 17.

Table 16: Performance of models augmented with knowledge bank via retrievers. *Oracle* indicates using *ground-truth* terms. Retri. Acc is short for retriever's accuracy score.

| Setting | *Statement judging* | | | | | *Multi-choice question answering* | | | | |
|---|---|---|---|---|---|---|---|---|---|---|
| | Retr. Acc | gpt-4o | gpt-4o -mini | Llama- 3.1-405B | Llama- 3.1-8B | Retr. Acc | gpt-4o | gpt-4o -mini | Llama- 3.1-405B | Llama- 3.1-8B |
| w.o. knowledge | 0.0 | 84.0 | 76.5 | 83.6 | 65.3 | 0.0 | 91.5 | 86.8 | 91.9 | 77.8 |
| BM25 | 34.6 | 86.5 ↑ 2.5 | 80.7 ↑ 4.2 | 83.9 ↑ 0.3 | 69.2 ↑ 3.9 | 29.7 | 92.3 ↑ 0.8 | 89.7 ↑ 2.9 | 90.8 ↓ 1.1 | 80.8 ↑ 3.0 |
| Ada Embed. | 41.2 | 85.9 ↑ 1.9 | 79.6 ↑ 3.1 | 86.0 ↑ 2.4 | 69.6 ↑ 4.3 | 47.9 | 92.1 ↑ 0.6 | 90.0 ↑ 3.2 | 92.0 ↑ 0.1 | 82.3 ↑ 4.5 |
| Oracle | 100.0 | 85.7 ↑ 1.7 | 81.1 ↑ 4.6 | 85.6 ↑ 2.0 | 69.2 ↑ 3.9 | 100.0 | 93.8 ↑ 2.3 | 90.0 ↑ 3.2 | 93.4 ↑ 1.5 | 81.6 ↑ 3.8 |

| Setting | *Financial calculation* | | | | | *All* | | | | |
|---|---|---|---|---|---|---|---|---|---|---|
| | Retr. Acc | gpt-4o | gpt-4o -mini | Llama- 3.1-405B | Llama- 3.1-8B | Retr. Acc | gpt-4o | gpt-4o -mini | Llama- 3.1-405B | Llama- 3.1-8B |
| w.o. knowledge | 0.0 | 31.8 | 26.5 | 28.1 | 12.8 | 0.0 | 63.6 | 57.4 | 61.9 | 45.5 |
| BM25 | 26.8 | 31.3 ↓ 0.5 | 27.0 ↑ 0.5 | 27.8 ↓ 0.3 | 13.4 ↑ 0.6 | 30.6 | 64.6 ↑ 1.0 | 59.9 ↑ 2.5 | 61.8 ↓ 0.1 | 47.9 ↑ 2.4 |
| Ada Embed. | 35.3 | 32.0 ↑ 0.2 | 26.3 ↓ 0.2 | 26.2 ↓ 1.9 | 14.2 ↑ 1.4 | 39.8 | 64.6 ↑ 1.0 | 59.2 ↑ 1.8 | 62.2 ↑ 0.3 | 48.6 ↑ 3.1 |
| Oracle | 100.0 | 33.0 ↑ 1.2 | 27.1 ↑ 0.6 | 30.3 ↑ 2.2 | 14.5 ↑ 1.7 | 100.0 | 65.2 ↑ 1.6 | 60.2 ↑ 2.8 | 64.0 ↑ 2.0 | 48.5 ↑ 3.0 |

Table 17: Performance of models augmented with knowledge bank across five capabilities for complex finance problem solving. *Oracle* indicates using *ground-truth* terms. Retri. Acc is short for retriever's accuracy score.

| Setting | *Terminology understanding* | | | | *Temporal reasoning* | | | |
|---|---|---|---|---|---|---|---|---|
| | gpt-4o | gpt-4o -mini | Llama- 3.1-405B | Llama- 3.1-8B | gpt-4o | gpt-4o -mini | Llama- 3.1-405B | Llama- 3.1-8B |
| w.o. knowledge | 85.4 | 78.4 | 85.3 | 68.0 | 24.6 | 19.9 | 16.1 | 7.9 |
| BM25 | 87.5 | 82.4 | 85.3 | 71.7 | 23.9 | 18.5 | 14.4 | 6.1 |
| Ada Embed. | 87.3 | 81.6 | 84.8 | 72.2 | 23.9 | 19.2 | 14.3 | 7.4 |
| Oracle | 87.4 | 82.9 | 87.9 | 71.9 | 24.6 | 20.8 | 17.0 | 10.0 |

| Setting | *Future forecasting* | | | | *Scenario planning* | | | |
|---|---|---|---|---|---|---|---|---|
| | gpt-4o | gpt-4o -mini | Llama- 3.1-405B | Llama- 3.1-8B | gpt-4o | gpt-4o -mini | Llama- 3.1-405B | Llama- 3.1-8B |
| w.o. knowledge | 63.6 | 58.6 | 70.5 | 50.5 | 38.6 | 33.7 | 34.5 | 18.9 |
| BM25 | 64.8 | 60.5 | 75.8 | 50.5 | 37.8 | 35.4 | 32.4 | 18.5 |
| Ada Embed. | 63.6 | 58.0 | 71.6 | 54.7 | 38.2 | 35.8 | 26.5 | 21.0 |
| Oracle | 61.1 | 59.3 | 73.7 | 51.6 | 38.2 | 35.8 | 32.4 | 20.6 |

| Setting | *modelling* | | | | *All* | | | |
|---|---|---|---|---|---|---|---|---|
| | gpt-4o | gpt-4o -mini | Llama- 3.1-405B | Llama- 3.1-8B | gpt-4o | gpt-4o -mini | Llama- 3.1-405B | Llama- 3.1-8B |
| w.o. knowledge | 42.0 | 35.7 | 33.8 | 19.2 | 63.6 | 57.4 | 61.9 | 45.5 |
| BM25 | 41.3 | 37.3 | 33.5 | 17.5 | 64.6 | 59.9 | 61.8 | 47.9 |
| Ada Embed. | 42.0 | 36.4 | 34.4 | 17.2 | 64.6 | 59.2 | 62.2 | 48.6 |
| Oracle | 42.5 | 34.5 | 36.7 | 21.0 | 65.2 | 60.2 | 64 | 48.5 |

# F MORE ERROR ANALYSIS

## F.1 HUMAN LABELING GUIDELINE

For errors in *financial calculation* task, we sampled 400 responses of o1 and assign them to three annotators. Our annotators are asked to determine 1) whether the reasoning path of o1's response coherets with the gold answer of corresponding correct answer; 2) whether there is rounding error in the intermediate calculating steps, *i.e.*, *rounding error*; and 3) whether the formula in o1's response is different from the formulas in the relevant finance terms, *i.e.*, *formula misuse*. During this process, annotators are provided with the gold answer of the corresponding after-class questions, which include the correct reasoning path. The result of each dimension is decided by at least two annotator's agreement.

For errors in visual-context *multi-choice question answering* task, we sampled 100 responses of GPT-4o and assign them to three annotators. Our annotators are asked to determine the explanation in gpt-4o's response is totally correct, partially correct, or wrong (Lu et al., 2024). For responses with partially correct and wrong explanation, we further ask annotators to decide 1) if the response presents correct reasoning path with consistency and correct interpretation of visual context; 2) if the response shows the model has difficulty identifying the positions and intersections of curves, *i.e.*, *blindness*; and 3) if the response misuses financial knowledge that leads to the error in the following reasoning steps, *i.e.*, *knowledge misuse*. The result of each dimension is decided by at least two annotator's agreement.

For errors in knowledge augmentation method, we sampled 100 responses of gpt-4o that give wrong final answers, and assign them to three annotators. Our annotators are asked to determine 1) whether the first wrong reasoning step is triggered by the information in the augmented knowledge (reasoning error if no); 2) whether the augmented knowledge proposes direct solution or evidence to answer the corresponding question (over thinking if yes); and 3) whether the wrong reasoning path is led by following every detail in the augmented knowledge (over reliance if yes). During this process, annotators are provided with the gold answer of the corresponding after-class questions, which include the correct reasoning path. The result of each dimension is decided by at least two annotator's agreement.

## F.2   ERROR CASES OF VISUAL CONTEXT

Figure 9 showcases two examples of responses from gpt-4o that come with visual context. In Figure 9(b), gpt-4o does not correctly apply finance knowledge and fails to identify the exact intersections of curves, while it still output the correct answer.

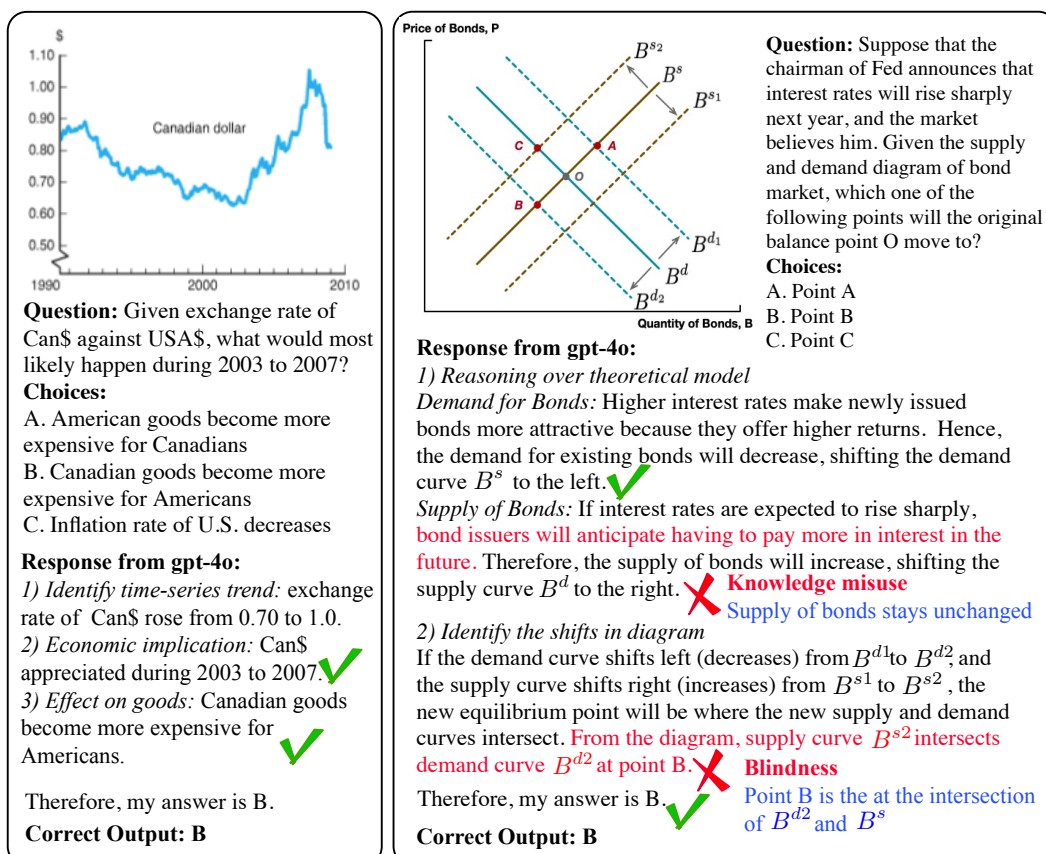

(a) Correct Ans. w. Correct Exp.

(b) Correct Ans. w. Wrong Exp.

Figure 9: Two examples from gpt-4o. In (b), gpt-4o does not correctly apply finance knowledge and fails to identify the exact intersections of curves, while it still output the correct answer.

## F.3   ERROR CASES OF KNOWLEDGE AUGMENTATION

The following three boxes present three examples that demonstrate the three error types in knowledge augmentation method, *i.e.*, Reasoning Error, Over Reliance, and Over Thinking, respectively.

Example of reasoning error in GPT-4o's response in *Oracle* setting. Red sentences are related to the wrong reasoning step.

Question (Financial Calculation task; Numerical Reasoning capability):
The Huff Co. has just gone public. Under a firm commitment agreement, Huff received $15.05 for each of the 5 million shares sold. The initial offering price was $16 per share, and the stock rose to $19.50 per share in the first few minutes of trading. Huff paid $800,000 in direct legal and other costs and $250,000 in indirect costs. What were the total indirect costs incurred by the company? (Unit: dollar)

Term and Definition:
Indirect expenses,Flotation costs. Indirect expenses are not reported on the prospectus and include the costs of management time spent working on the new issue.

Model Response:
To determine the total indirect costs incurred by Huff Co., we only need to consider the provided information about the indirect expenses. The question states that Huff Co. incurred $250,000 in indirect costs. Since no other indirect costs are mentioned, these will be the total indirect costs. Therefore, my answer is [250000].

Correct Answer:
17750000

Analysis:
The wrong reasoning path is not triggered by the information in the augmented knowledge.

---

Example of over reliance error in GPT-4o's response in *Oracle* setting. Red sentences are related to the wrong reasoning step while teal sentences enhance or correct the model's reasoning process.

Question (Financial Calculation task; Scenario Planning capability):
A stock price is currently $40. At the end of one month, it will be either $42 or $38. The risk-free interest rate is 8% per annum with continuous compounding. What is the value of a one-month European call option with a strike price of $39? (Unit: dollar, decimal)

Term and Definition:
One-step,Binomial tree. A no-arbitrage framework for pricing options on a stock is presented, where the stock price is denoted as $S_0$ and the current option price as $f$. ...... Rearranging provides: $f = S_0\Delta(1 - ue^{-rT}) + f_ue^{-rT}$. ...... This can be simplified to: $f = \frac{f_u(1-de^{-rT})+f_d(ue^{-rT}-1)}{u-d}$. We also introduce $p$ defined as: $p = \frac{e^{rT}-d}{u-d}$.

Model Response:
To determine the value of a one-month European call option with a strike price of $39, we will use the provided binomial tree model framework along with the relevant equations. Step-by-Step Solution ...... Determine the value of $p$ using provided Equation: $p = \frac{e^{rT}-d}{u-d} = \frac{1.0067-0.95}{1.05-0.95} = \frac{0.0567}{0.1} = 0.567$. Calculate the value of the call option using another provided Equation: $f = \frac{f_u(1-de^{-rT})+f_d(ue^{-rT}-1)}{u-d} = 1.70595$

Correct Answer:
1.69

Analysis:
The last step calculation that leads to the wrong final answer is not necessarily so complex. The model could simply solve this problem via planning in different scenarios. The expected payoff $E$ is given by: $E = p \cdot 3 + (1 - p) \cdot 0 = 1.701$. The value of the call option $C$ is the discounted expected payoff: $C = E \cdot e^{-rT} = 1.701 \cdot 0.9934 \approx 1.69$.

Example of over thinking error in GPT-4o's response in *Oracle* setting. Red sentences are related to the wrong reasoning step while teal sentences enhance or correct the model's reasoning process.

Question (Statement Judging task; Future Forecasting capability):
Statement: Consider the operating procedures of the Fed in response to movements in output. If the Fed has interest-rate targets, a rise in output that raises interest rates might cause the Fed to sell bonds and bid down their price.

Term and Definition:
Traditional Interest-Rate Channels,Transmission Mechanisms of Monetary Policy. The traditional view of the monetary transmission mechanism illustrates how expansionary monetary policy impacts the economy through a schematic representation: $\Rightarrow i_t \downarrow \Rightarrow l \uparrow \Rightarrow Y \uparrow$. An expansionary monetary policy results in a decrease in real interest rates ($i_t, \downarrow$), which lowers the cost of capital, thus increasing investment spending ($l \uparrow$) and boosting aggregate demand, leading to a rise in output ($Y \uparrow$). Under interest-rate target, the government could hold back the changes on interest rate via open market operations. For example, selling bonds would cut down the interest rate due to the increasing money supply.

Model Response:
Let's break down the given statement and consider ......3. Fed's Response with Interest-Rate Targeting: If the Fed observes a rise in interest rates due to increased output but desires to maintain its interest-rate target, it would engage in open market operations to influence these rates. 4.Selling Bonds to Lower Prices and Raise Rates: The Fed sells bonds in open market operations to reduce the money supply. Selling bonds bids down their price, which inversely raises the yield or interest rates. This action can counter rising rates if the objective is to maintain higher interest rates to dampen potential inflationary pressures. Putting it together...... Therefore, my answer is true.

Correct Answer:
No

Analysis:
After the third reasoning step, the correct answer could be proposed. However, the model thinks one step further to consider other influencing factors in the long run, and finally give the wrong final answer. Note that the third reasoning step highly follows the part of augmented knowledge highlighted in blue.

## G    PROMPT TEMPLATES

### G.1    SENSITIVITY ANALYSIS

We conduct sensitivity analysis on prompt templates for evaluation on XFINBENCH. ProSA (Zhuo et al., 2024) showcases four different styles of constructing prompts, *i.e.*, *simple input* (SI), *emotional support* (ES), *role player* (RP) and *output requirement* (OR). We further include two common prompting strategies, *i.e.*, *chain-of-though* (CoT) and *direct answering* (DA). Hence, we design four types of prompt templates for conducting our sensitivity analysis, as shown in Table 18. Note that *output requirement* is indispensable in our tasks for automatic evaluating the model's final answers.

Table 18: Four prompt templates for sensitivity analysis during evaluation.

| Capability | Task |
|---|---|
| CoT & RP & OR | You are a financial expert. You are supposed to answer the given question.\n Question: {after-class question}\n Please answer the above question and output your final answer starting with 'Therefore, my answer is' at the end, where you store you final answer into '[]'.\n Let's think step by step.\n |
| DA & RP & OR | You are a financial expert. You are supposed to answer the given question.\n Question: {after-class question}\n Please answer the above question and output your final answer starting with 'Therefore, my answer is' at the end, where you store you final answer into '[]'.\n |
| CoT & OR | Question: {after-class question}\n Please answer the above question and output your final answer starting with 'Therefore, my answer is' at the end, where you store you final answer into '[]'.\n Let's think step by step.\n |
| DA & OR | Question: {after-class question}\n Please answer the above question and output your final answer starting with 'Therefore, my answer is' at the end, where you store you final answer into '[]'.\n |

We randomly sample 500 examples from the test set of XFINBENCH and use them to evaluate four models on each of prompt templates in Table 18. Experiment results in Table 19 show that the prompt template involving *chain-of-though*, *role player* and *output requirement* consistently leads to outstanding performance of most models across three tasks, and brings out the best performance of most models with slight margins. Additionally, *Chain-of-thought* strategy outperforms *direct answering* strategy under most cases since our financial tasks require intensive mathematical and logical reasoning (Sprague et al., 2024). Despite the slight differences of performance across four prompt templates, the rankings of four models hardly change in three tasks and the overall scores.

Table 19: Performance of models using different prompt templates during evaluation.

| Setting | Statement judging | | | | Multiple-choice question | | | |
|---|---|---|---|---|---|---|---|---|
| | gpt-4o | gpt-4o -mini | Llama- 3.1-405B | Llama- 3.1-8B | gpt-4o | gpt-4o -mini | Llama- 3.1-405B | Llama- 3.1-8B |
| CoT & RP & OR | 80.6 | 71.8 | 77.8 | 62.8 | 89.1 | 79.1 | 83.6 | 62.7 |
| DA & RP & OR | 80.6 | 65.0 | 76.1 | 58.3 | 88.1 | 74.6 | 83.6 | 68.7 |
| CoT & OR | 82.2 | 72.2 | 77.8 | 55.6 | 88.1 | 74.6 | 85.1 | 65.7 |
| DA & OR | 76.7 | 65.0 | 77.8 | 58.3 | 89.6 | 71.6 | 85.1 | 61.2 |

| Setting | Financial calculation | | | | All | | | |
|---|---|---|---|---|---|---|---|---|
| | gpt-4o | gpt-4o -mini | Llama- 3.1-405B | Llama- 3.1-8B | gpt-4o | gpt-4o -mini | Llama- 3.1-405B | Llama- 3.1-8B |
| CoT & RP & OR | 31.0 / 52.0 | 21.7 / 36.8 | 18.2 / 30.4 | 8.3 / 16.2 | 56.6 | 47.5 | 48.4 | 35.2 |
| DA & RP & OR | 30.0 / 48.2 | 22.5 / 37.2 | 16.2 / 28.5 | 9.5 / 15.8 | 56.0 | 44.8 | 46.8 | 35.0 |
| CoT & OR | 27.3 / 45.5 | 21.3 / 34.4 | 20.6 / 33.2 | 6.7 / 11.9 | 55.2 | 46.8 | 49.8 | 32.2 |
| DA & OR | 27.3 / 46.2 | 19.4 / 33.6 | 20.2 / 35.2 | 8.7 / 15.0 | 53.4 | 42.8 | 49.6 | 33.6 |

## G.2 PROMPT FOR DATASET CONSTRUCTION

We apply the generate-then-verify paradigm for constructing our dataset. Prompts used in the generate-then-verify paradigm for *statement judging*, *multi-choice question answering*, and *financial calculation* tasks, are shown in G.2.1, G.2.2, and G.2.3, respectively. Additionally, we use the prompt in G.2.4 to annotate the calculation problems in our dataset with mathematical reasoning abilities summarized in Lu et al. (2024).

### G.2.1 PROMPT FOR STATEMENT JUDGING TASK

---

**Prompt template for generating true statements in statement judging task.**

Please rewrite a question-answer pair into one or more statement(s) which is/are true. Specifically,

1. The statement(s) should be generated from the original question-answer pair and must be true given the content of the question-answer pair.

2. None of the following expressions is allowed in the statement: (1) unclear pronoun; (2) in/given/according to the chapter/figure/table; (3) conjunctions of causality like since, because and so on.

3. You should extract the context of the original question. The context usually introduces the background of the generated statement(s). Note that: (1) the context must NOT be question; (2) there should NOT be duplicated or contradictory information between the context and the statement.

4. You are allowed to generate two or more statements from one question-answer pair. Under this case, the statements should be independent of each other, with as little overlap as possible.

Example 1:
{example 1}

Example 2:
{example 2}

Example 3:
{example 3}

Example 4:
{example 4}

Example 5:
{example 5}

Given the above instructions and examples, please use the following question-answer pair to generate at least one statement with a clear answer and context.
Original Question: {after-class question}
Original Answer: {after-class solution}

---

> **Example 1 in prompt template for generating true statements in statement judging task.**
>
> Original Question: Suppose that a bond portfolio with a duration of 12 years is hedged using a futures contract in which the underlying asset has a duration of four years. What is likely to be the impact on the hedge of the fact that the 12-year rate is less volatile than the four-year rate?
>
> Original Answer: Duration-based hedging procedures assume parallel shifts in the yield curve. Since the 12-year rate tends to move by less than the 4-year rate, the portfolio manager may find that he or she is over-hedged.
>
> Context: Suppose that a bond portfolio with a duration of 12 years is hedged using a futures contract in which the underlying asset has a duration of four years.
>
> Statement: Considering duration-based hedging procedures assume parallel shifts in the yield curve, the portfolio manager may find that he or she is over-hedged.
>
> Answer: True

> **Example 2 in prompt template for generating true statements in statement judging task.**
>
> Original Question: What is meant by the delta of a stock option?
>
> Original Answer: The delta of a stock option measures the sensitivity of the option price to the price of the stock when small changes are considered. Specifically, it is the ratio of the change in the price of the stock option to the change in the price of the underlying stock.
>
> Context: Consider the delta of a stock option.
>
> Statement: The delta of a stock option measures the sensitivity of the option price to the price of the stock when small changes are considered.
>
> Answer: True
>
> Statement: The delta of a stock option is the ratio of the change in the price of the stock option to the change in the price of the underlying stock.
>
> Answer: True

**Example 3 in prompt template for generating true statements in statement judging task.**

Original Question: The treasurer of a corporation is trying to choose between options and forward contracts to hedge the corporation's foreign exchange risk. Discuss the advantages and disadvantages of each.

Original Answer: Forward contracts lock in the exchange rate that will apply to a particular transaction in the future. Options provide insurance that the exchange rate will not be worse than some level. The advantage of a forward contract is that uncertainty is eliminated as far as possible. The disadvantage is that the outcome with hedging can be significantly worse than the outcome with no hedging. This disadvantage is not as marked with options. However, unlike forward contracts, options involve an up-front cost.

Context: The treasurer of a corporation is trying to choose between options and forward contracts to hedge the corporation's foreign exchange risk.

Statement: Options provide insurance that the exchange rate will not be worse than some level.
Answer: True

Statement: When using forward contracts for hedging, the outcome can be worse than the outcome with no hedging.
Answer: True

Statement: Using options to hedge the foreign exchange risk involve an up-front cost.
Answer: True

---

Example 4 in prompt template for generating true statements in statement judging task.

Original Question: The term structure of interest rates is upward sloping. Put the following in order of magnitude:
(a) The five-year zero rate
(b) The yield on a five-year coupon-bearing bond
(c) The forward rate corresponding to the period between 4.75 and 5 years in the future
What is the answer to this question when the term structure of interest rates is downward sloping?

Original Answer: When the term structure is upward sloping, $c > a > b$. When it is downward sloping, $b > a > c$.

Context: The term structure of interest rates is upward sloping.

Statement: The five-year zero rate is smaller than the forward rate corresponding to the period between 4.75 and 5 years in the future.
Answer: True

Statement: The yield on a five-year coupon-bearing bond is smaller than the forward rate corresponding to the period between 4.75 and 5 years in the future.
Answer: True

Statement: The yield on a five-year coupon-bearing bond is larger than the five-year zero rate.
Answer: True

Statement: The five-year zero rate is larger than the forward rate corresponding to the period between 4.75 and 5 years in the future.
Answer: True

---

Example 5 in prompt template for generating true statements in statement judging task.

Original Question: For each of the following scenarios, discuss whether profit opportunities exist from trading in the stock of the firm under the conditions that (1) the market is not weak form efficient, (2) the market is weak form but not semistrong form efficient, (3) the market is semistrong form but not strong form efficient, and (4) the market is strong form efficient. **a.** The stock price has risen steadily each day for the past 30 days. **b.** The financial statements for a company were released three days ago, and you believe you've uncovered some anomalies in the company's inventory and cost control reporting techniques that are causing the firm's true liquidity strength to be understated. **c.** You observe that the senior managers of a company have been buying a lot of the company's stock on the open market over the past week.

Original Answer:
(a). If the market is not weak form efficient, then this information could be acted on and a profit earned from following the price trend. Under (2), (3), and (4), this information is fully impounded in the current price and no abnormal profit opportunity exists.
(b). Under (2), if the market is not semi-strong form efficient, then this information could be used to buy the stock c̈heapb̈efore the rest of the market discovers the financial statement anomaly. Since (2) is stronger than (1), both imply that a profit opportunity exists; under (3) and (4), this information is fully impounded in the current price and no profit opportunity exists.
(c). Under (3), if the market is not strong form efficient, then this information could be used as a profitable trading strategy, by noting the buying activity of the insiders as a signal that the stock is underpriced or that good news is imminent. Since (1) and (2) are weaker than (3), all three imply that a profit opportunity exists. Note that this assumes the individual who sees the insider trading is the only one who sees the trading. If the information about the trades made by company management is public information, it will be discounted in the stock price and no profit opportunity exists. Under (4), this information does not signal any profit opportunity for traders; any pertinent information the manager-insiders may have is fully reflected in the current share price.

Context: Consider profit opportunities exist from trading in the stock of the firm.

Statement: In a market that is not weak form efficient, a profit could be earned from acting on the information of a stock price that has risen steadily each day for the past 30 days.
Answer: True

Statement: In a market that is not semi-strong form efficient, a profit could be earned from acting on the pertinent information the manager-insiders may have.
Answer: True

Statement: In a market that is not strong form efficient, there is no profit opportunity on the information that you observe that the senior managers of a company have been buying a lot of the company's stock on the open market over the past week.
Answer: True

---

**Prompt template for generating false statements in statement judging task.**

Please rewrite a question-answer pair into one or more statement(s) which is/are false. Specifically,

1. The statement(s) should be generated from the original question-answer pair and must be false given the content of the question-answer pair.

2. None of the following expressions is allowed in the statement: (1) unclear pronoun; (2) in/given/according to the chapter/figure/table; (3) conjunctions of causality like since, because and so on.

3. You should extract the context of the original question. The context usually introduces the background of the generated statement(s). Note that: (1) the context must NOT be question; (2) there should NOT be duplicated or contradictory information between the context and the statement.

4. You are allowed to generate two or more statements from one question-answer pair. Under this case, the statements should be independent of each other, with as little overlap as possible.

Example 1:
{example 1}

Example 2:
{example 2}

Example 3:
{example 3}

Example 4:
{example 4}

Example 5:
{example 5}

Given the above instructions and examples, please use the following question-answer pair to generate at least one statement with a clear answer and context.
Original Question: {after-class question}
Original Answer: {after-class solution}

---

**Example 1 in prompt template for generating false statements in statement judging task.**

Original Question: Suppose that a bond portfolio with a duration of 12 years is hedged using a futures contract in which the underlying asset has a duration of four years. What is likely to be the impact on the hedge of the fact that the 12-year rate is less volatile than the four-year rate?

Original Answer: Duration-based hedging procedures assume parallel shifts in the yield curve. Since the 12-year rate tends to move by less than the 4-year rate, the portfolio manager may find that he or she is over-hedged.

Context: Suppose that a bond portfolio with a duration of 12 years is hedged using a futures contract in which the underlying asset has a duration of four years.

Statement: Considering duration-based hedging procedures assume parallel shifts in the yield curve, the portfolio manager may find that he or she is under-hedged.
Answer: False

---

**Example 2 in prompt template for generating false statements in statement judging task.**

Original Question: What is meant by the delta of a stock option?

Original Answer: The delta of a stock option measures the sensitivity of the option price to the price of the stock when small changes are considered. Specifically, it is the ratio of the change in the price of the stock option to the change in the price of the underlying stock.

Context: Consider the delta of a stock option.

Statement: The delta of a stock option measures the sensitivity of the option price to the price of the stock when big changes are considered.
Answer: False

**Example 3 in prompt template for generating false statements in statement judging task.**

Original Question: The treasurer of a corporation is trying to choose between options and forward contracts to hedge the corporation's foreign exchange risk. Discuss the advantages and disadvantages of each.

Original Answer: Forward contracts lock in the exchange rate that will apply to a particular transaction in the future. Options provide insurance that the exchange rate will not be worse than some level. The advantage of a forward contract is that uncertainty is eliminated as far as possible. The disadvantage is that the outcome with hedging can be significantly worse than the outcome with no hedging. This disadvantage is not as marked with options. However, unlike forward contracts, options involve an up-front cost.

Context: The treasurer of a corporation is trying to choose between options and forward contracts to hedge the corporation's foreign exchange risk.

Statement: When using forward contracts for hedging, the outcome is definitely better than the outcome with no hedging.
Answer: False

Statement: Using forward contracts to hedge the foreign exchange risk involve an up-front cost.
Answer: False

---

**Example 4 in prompt template for generating false statements in statement judging task.**

Original Question: The term structure of interest rates is upward sloping. Put the following in order of magnitude:
(a) The five-year zero rate
(b) The yield on a five-year coupon-bearing bond
(c) The forward rate corresponding to the period between 4.75 and 5 years in the future
What is the answer to this question when the term structure of interest rates is downward sloping?

Original Answer: When the term structure is upward sloping, $c > a > b$. When it is downward sloping, $b > a > c$.

Context: The term structure of interest rates is upward sloping.

Statement: The five-year zero rate is larger than the forward rate corresponding to the period between 4.75 and 5 years in the future.
Answer: False

Statement: The yield on a five-year coupon-bearing bond is larger than the forward rate corresponding to the period between 4.75 and 5 years in the future.
Answer: False

Statement: When it is downward sloping, the yield on a five-year coupon-bearing bond is smaller than the five-year zero rate.
Answer: False

Statement: When it is downward sloping, The five-year zero rate is smaller than the forward rate corresponding to the period between 4.75 and 5 years in the future.
Answer: False

> **Example 5 in prompt template for generating false statements in statement judging task.**
>
> Original Question: For each of the following scenarios, discuss whether profit opportunities exist from trading in the stock of the firm under the conditions that (1) the market is not weak form efficient, (2) the market is weak form but not semistrong form efficient, (3) the market is semistrong form but not strong form efficient, and (4) the market is strong form efficient. **a.** The stock price has risen steadily each day for the past 30 days. **b.** The financial statements for a company were released three days ago, and you believe you've uncovered some anomalies in the company's inventory and cost control reporting techniques that are causing the firm's true liquidity strength to be understated. **c.** You observe that the senior managers of a company have been buying a lot of the company's stock on the open market over the past week.
>
> Original Answer:
> (a). If the market is not weak form efficient, then this information could be acted on and a profit earned from following the price trend. Under (2), (3), and (4), this information is fully impounded in the current price and no abnormal profit opportunity exists.
> (b). Under (2), if the market is not semi-strong form efficient, then this information could be used to buy the stock c̈heap̈before the rest of the market discovers the financial statement anomaly. Since (2) is stronger than (1), both imply that a profit opportunity exists; under (3) and (4), this information is fully impounded in the current price and no profit opportunity exists.
> (c). Under (3), if the market is not strong form efficient, then this information could be used as a profitable trading strategy, by noting the buying activity of the insiders as a signal that the stock is underpriced or that good news is imminent. Since (1) and (2) are weaker than (3), all three imply that a profit opportunity exists. Note that this assumes the individual who sees the insider trading is the only one who sees the trading. If the information about the trades made by company management is public information, it will be discounted in the stock price and no profit opportunity exists. Under (4), this information does not signal any profit opportunity for traders; any pertinent information the manager-insiders may have is fully reflected in the current share price.
>
> Context: Consider profit opportunities exist from trading in the stock of the firm.
>
> Statement: In a market that is weak form efficient but not semistrong form efficient, a profit could be earned from acting on the information of a stock price that has risen steadily each day for the past 30 days.
> Answer: False
>
> Statement: In a market that is strong form efficient, a profit could be earned from acting on the pertinent information the manager-insiders may have.
> Answer: False
>
> Statement: In a market that is semistrong form but not strong form efficient, there is no profit opportunity on the information that you observe that the senior managers of a company have been buying a lot of the company's stock on the open market over the past week.
> Answer: False

---

**Prompt template for verifying true statements in *statement judging* task.**

Original Question: {`after-class question`}
Original Answer: {`after-class solution`}
Context of Statement: {`context`}
Statement: {`question`}
Given the above original question and answer, please answer the following two questions.
Q1: Is the statement definitely true given the original question and answer?
Q2: Does the context extract the essential background information in the original question?
Your Answer to Q1 and Q2 (Yes or No, no explanation required):

---

**Prompt template for verifying false statements in *statement judging* task.**

Original Question: {`after-class question`}
Original Answer: {`after-class solution`}
Context of Statement: {`context`}
Statement: {`question`}
Given the above original question and answer, please answer the following two questions.
Q1: Is the statement definitely false given the original question and answer?
Q2: Does the context extract the essential background information in the original question?
Your Answer to Q1 and Q2 (Yes or No, no explanation required):

---

**Prompt template for deduplicating dependent statements in *statement judging* task.**

Context of Statements: {`context`}
Statement 1: {`true statement`}
Statement 2: {`false statement`}
Please determine whether Statement 1 provides direct evidence to support that Statement 2 is false.
Your Answer (Yes or No):

### G.2.2 PROMPT FOR MULTI-CHOICE QUESTION ANSWERING TASK

---

**Prompt template for generating questions in multi-choice question answering task.**

Please rewrite a question-answer pair into one or more question(s) with three candidate choices. Specifically,

1. The question and correct answer should be generated from the question and/or answer, under a clear and concise wording style. None of the following expressions is allowed in the question: (1) unclear pronoun; (2) in/given/according to the chapter/figure/table.

2. There are three candidate choices for the question. The correct answer lies in Choice (a), and Choice (b) and (c) are both wrong to the question. Choice (a), (b) and (c), should be independent and mutually exclusive. Noising choices, i.e. (b) and (c), should share the similar wording and length with the correct answer (a). Choice (b) reflects a misunderstanding of the original question-answer pair, while Choice (c) is made up by you.

3. You should extract the context of the original question. The context usually introduces the background of the generated question(s). Note that: (1) the context must NOT be question; (2) there should NOT be duplicated or contradictory information between the context and the statement.

4. You are allowed to generate two or more questions from one original question-answer pair. Under this case, the questions should be independent of each other, with as little overlap as possible.

Example 1:
{example 1}

Example 2:
{example 2}

Example 3:
{example 3}

Example 4:
{example 4}

Example 5:
{example 5}

Given the above instructions and examples, please use the following question-answer pair to generate at least one question with candidate choices and context.
Original Question: {after-class question}
Original Answer: {after-class solution}

---

> **Example 1 in prompt template for generating questions in multi-choice question answering task.**
>
> Original Question: Last month, BlueSky Airline announced that it would stretch out its bill payments to 45 days from 30 days. The reason given was that the company wanted to çontrol costs and optimize cash flow.Ṫhe increased payables period will be in effect for all of the company's 4,000 suppliers. Why don't all firms simply increase their payables periods to shorten their cash cycles?
>
> Original Answer: They would like to! The payables period is a subject of much negotiation, and it is one aspect of the price a firm pays its suppliers. A firm will generally negotiate the best possible combination of payables period and price. Typically, suppliers provide strong financial incentives for rapid payment. This issue is discussed in detail in a later chapter on credit policy.
>
> Context: Last month, BlueSky Airline announced that it would stretch out its bill payments to 45 days from 30 days.
>
> Generated Question: Which one of the following choices is one of the reasons of BlueSky Airline announcement?
> Choices:
> (a) Optimize cash flow
> (b) Increase investment in fixed costs
> (c) Increase sales volume
> Correct Answer: a

> **Example 2 in prompt template for generating questions in multi-choice question answering task.**
>
> Original Question: What are the advantages of using the DCF model for determining the cost of equity capital? What are the disadvantages? What specific piece of information do you need to find the cost of equity using this model? What are some of the ways in which you could get this estimate?
>
> Original Answer: The primary advantage of the DCF model is its simplicity. The method is disadvantaged in that (1) the model is applicable only to firms that actually pay dividends; many do not; (2) even if a firm does pay dividends, the DCF model requires a constant dividend growth rate forever; (3) the estimated cost of equity from this method is very sensitive to changes in g, which is a very uncertain parameter; and (4) the model does not explicitly consider risk, although risk is implicitly considered to the extent that the market has impounded the relevant risk of the stock into its market price. While the share price and most recent dividend can be observed in the market, the dividend growth rate must be estimated. Two common methods of estimating g are to use analysts' earnings and payout forecasts or to determine some appropriate average historical g from the firm's available data.
>
> Context: The DCF model have advantages and disadvantages for determining the cost of equity capital.
>
> Generated Question: Which one of the following advantages do the DCF model have?
> Choices:
> (a) Simple calculation
> (b) Applicable for firms that do not pay dividends
> (c) Insensitivity to the financial environment
> Correct Answer: a

**Example 3 in prompt template for generating questions in multi-choice question answering task.**

Original Question: 'When a bank is negotiating currency swaps, it should try to ensure that it is receiving the lower interest rate currency from a company with a low credit risk.' Explain.

Original Answer: As time passes there is a tendency for the currency which has the lower interest rate to strengthen. This means that a swap where we are receiving this currency will tend to move in the money (i.e., have a positive value). Similarly a swap where we are paying the currency will tend to move out of the money (i.e., have a negative value). From this it follows that our expected exposure on the swap where we are receiving the low-interest currency is much greater than our expected exposure on the swap where we are receiving the high-interest currency. We should therefore look for counterparties with a low credit risk on the side of the swap where we are receiving the low-interest currency. On the other side of the swap we are far less concerned about the creditworthiness of the counterparty.

Context: A bank is negotiating currency swaps.

Generated Question: Which one of the following actions should it consider?
Choices:
(a) Seek counterparties with low credit risk where the bank is receiving the low-interest currency
(b) Seek counterparties with high credit risk where the bank is receiving the low-interest currency
(c) Seek counterparties with low credit risk where the bank is receiving the high-interest currency
Correct Answer: a

> **Example 4 in prompt template for generating questions in multi-choice question answering task.**
>
> Original Question: How can bank behavior and the Fed's behavior cause money supply growth to be precyclical (rising in booms and falling in recessions)?
>
> Original Answer: Bank behavior can lead to procyclical money growth because when interest rates rise in a boom, they decrease excess reserves and increase their borrowing from the Fed, both of which lead to a higher money supply. Similarly, when interest rates fall in a recession, they increase excess reserves and decrease their borrowing from the Fed, leading to a lower money supply. The result is that the money supply will tend to grow faster in booms and slower in recessions–it is procyclical. Fed behavior also can lead to procyclical money growth because (as the answer to problem 1 indicates) an interest-rate target can lead to a slower rate of growth of the money supply during recessions and a more rapid rate of growth during booms.
>
> Context: Bank behavior and the Fed's behavior can cause money supply growth to be precyclica.
>
> Generated Question: Which one of the following bank and/or the Fed's behaviours would happen when interest rates rise in a boom?
> Choices:
> (a) Banks increase their borrowings from the Fed
> (b) Banks increase excess reserves
> (c) The Fed's make positive announcements
> Correct Answer: a
>
> Generated Question: Which one of the following bank and/or the Fed's behaviours would happen when interest rates rise in a recession?
> Choices:
> (a) Banks decrease their borrowings from the Fed
> (b) Banks decrease excess reserves
> (c) The Fed's make positive announcements
> Correct Answer: a

**Example 5 in prompt template for generating questions in multi-choice question answering task.**

Original Question: Which regulatory agency has the primary responsibility for supervising the following categories of commercial banks? a. National banks; b. Bank holding companies; c. Non-Federal Reserve member state banks; d. Federal Reserve member state banks

Original Answer: (a) Office of the Comptroller of the Currency; (b) the Federal Reserve; (c) state banking authorities and the FDIC; (d) the Federal Reserve

Context: Regulatory agencies have the primary responsibility for supervising commercial banks.

Generated Question: Which one of the following agencies has the primary responsibility for supervising national banks?
Choices:
(a) Office of the Comptroller of the Currency
(b) state banking authorities
(c) the Bank of Settlement
Correct Answer: a

Generated Question: Which one of the following agencies has the primary responsibility for supervising bank holding companies?
Choices:
(a) the Federal Reserve
(b) Office of the Comptroller of the Currency
(c) the International Monetary Fund
Correct Answer: a

Generated Question: Which one of the following agencies has the primary responsibility for supervising non-Federal Reserve member state banks?
Choices:
(a) state banking authorities and the FDIC
(b) the Federal Reserve
(c) the National Credit Union Administration
Correct Answer: a

Generated Question: Which one of the following agencies has the primary responsibility for supervising Federal Reserve member state banks?
Choices:
(a) the Federal Reserve
(b) the FDIC
(c) Financial Stability Oversight Council
Correct Answer: a

> **Prompt template for verifying questions in *multi-choice question answering* task.**
>
> Original Question: {`after-class question`}
> Original Answer: {`after-class solution`}
> Context of Generated Question: {`context`}
> Generated Question: {`question`}
> Candidate Choices:{`choices`}
> Correct Answer: {`answer`}
>
> Given the above original question and answer, please answer the following two questions.
> Q1: Is the correct answer definitely true to the generated question?
> Q2: Are the other two misleading answers within candidate choices definitely false to the generated question?
> Q3: Are the three candidate choices mutually exclusive but sharing the similar wording and length with each other?
> Q4: Does the context extract the essential background information in the original question?
> Your Answer to Q1, Q2, Q3 and Q4 (Yes or No, no explanation required):

### G.2.3 PROMPT FOR FINANCIAL CALCULATION TASK

---

**Prompt template for generating questions in financial calculation task.**

Please rewrite a question-answer pair into one or more question(s) with clear answer(s). Specifically,

1. The question should be generated from the original question-answer pair and written in a clear and concise wording style. The question should clarify the unit for its answer at the end if any.

2. The answer MUST be pure numbers from the original answer without any symbol attached. Specifically, it should be in decimal form and have no special symbols like percent sign and currency symbols.

3. You should extract the context of the original question. The context usually contains the necessary details for calculation, and serves as the background of the generated question(s). Note that: (1) the context must NOT be question; (2) there should NOT be duplicated or contradictory information between the context and the statement.

4. You are allowed to generate two or more questions from one question-answer pair, each with a answer. Under this case, the questions should be independent of each other. It is not allowed that the answer to any questions is an intermediate step to other questions.

Example 1:
{example 1}

Example 2:
{example 2}

Example 3:
{example 3}

Example 4:
{example 4}

Example 5:
{example 5}

Given the above instructions and examples, please use the following question-answer pair to generate at least one question with a clear answer and context.
Original Question: {after-class question}
Original Answer: {after-class solution}

---

---

**Example 1 in prompt template for generating questions in financial calculation task.**

Original Question: A credit default swap requires a semiannual payment at the rate of 60 basis points per year. The principal is $300 million and the credit default swap is settled in cash. A default occurs after four years and two months, and the calculation agent estimates that the price of the cheapest deliverable bond is 40% of its face value shortly after the default. List the cash flows and their timing for the seller of the credit default swap.

Original Answer: The seller receives

$$300,000,000 \times 0.0060 \times 0.5 = \$900,000$$

at times 0.5, 1.0, 1.5, 2.0, 2.5, 3.0, 3.5, and 4.0 years. The seller also receives a final accrual payment of about
$$\$300,000(= \$300,000,000 \times 0.060 \times 2/12)$$
at the time of the default (4 years and two months). The seller pays

$$300,000,000 \times 0.6 = \$180,000,000$$

at the time of the default. (This does not consider day count conventions.)

Context: A credit default swap requires a semiannual payment at the rate of 60 basis points per year. The principal is $300 million and the credit default swap is settled in cash. A default occurs after four years and two months, and the calculation agent estimates that the price of the cheapest deliverable bond is 40% of its face value shortly after the default.

Generated Question: What is the cash paid by the seller at the time of the default? (Unit: dollar)
Answer: 180000000.00

---

**Example 2 in prompt template for generating questions in financial calculation task.**

Original Question: Calculate the price of a three-month American put option on a non-dividend-paying stock when the stock price is \$60, the strike price is \$60, the risk-free interest rate is 10% per annum, and the volatility is 45% per annum. Use a binomial tree with a time interval of one month.

Original Answer: In this case, $S_0 = 60$, $K = 60$, $r = 0.1$, $\sigma = 0.45$, $T = 0.25$, and $\Delta t = 0.0833$. Also

$$u = e^{\sigma/\Delta t} = e^{0.45\sqrt{0.0833}} = 1.1387$$

$$d = \frac{1}{u} = 0.8782$$

$$a = e^{r\Delta t} = e^{0.1 \cdot 0.0833} = 1.0084$$

$$p = \frac{a - d}{u - d} = 0.4998$$

$$1 - p = 0.5002$$

The output from DerivaGem for this example is shown in the Figure S21.1. The calculated price of the option is \$5.16.
Figure S21.1: Tree for Problem 21.2 Context: Here is a three-month American put option on a non-dividend-paying stock. Suppose the stock price is \$60, the strike price is \$60, the risk-free interest rate is 10% per annum, and the volatility is 45% per annum.

Generated Question: What is the price of this put option using a binomial tree with a time interval of one month?
Answer: 5.16

---

**Example 3 in prompt template for generating questions in financial calculation task.**

Original Question: You want to buy a new sports coupe for $61,800, and the finance office at the dealership has quoted you a 7.4 percent APR loan for 60 months to buy the car. What will your monthly payments be? What is the effective annual rate on this loan?

Original Answer: We first need to find the annuity payment. We have the PVA, the length of the annuity, and the interest rate. Using the PVA equation:

$$PVA = C([1 - [1/(1+r)]^t]r)$$

$$\$61,800 = C[1 - [1/[1 + (.074/12)]^{60}]](.074/12)]$$

Solving for the payment, we get:

$$C = \$61,800/50.02385 = \$1,235.41$$

To find the EAR, we use the EAR equation:

$$EAR = [1 + (APR/m)]^m - 1$$

$$EAR = [1 + (.074/12)]^{12} - 1 = .0766$$

Context: You want to buy a new sports coupe for $61,800, and the finance office at the dealership has quoted you a 7.4 percent APR loan for 60 months to buy the car.

Generated Question: What will your monthly payments be? (Unit: dollar)
Answer: 1235.41

Generated Question: What is the effective annual rate on this loan?
Answer: 0.0766

> **Example 4 in prompt template for generating questions in financial calculation task.**
>
> Original Question: What is the value of an investment that pays \$7,500 every other year forever, if the first payment occurs one year from today and the discount rate is 11 percent compounded daily? What is the value today if the first payment occurs four years from today?
>
> Original Answer: The cash flows in this problem occur every two years, so we need to find the effective two year rate. One way to find the effective two year rate is to use an equation similar to the EAR, except use the number of days in two years as the exponent. (We use the number of days in two years since it is daily compounding; if monthly compounding was assumed, we would use the number of months in two years.) So, the effective two-year interest rate is: Effective 2-year rate $= [1 + (.11/365)]^{[}365(2)] - 1 = .2460$
> We can use this interest rate to find the PV of the perpetuity. Doing so, we find: $PV = \$7,500/.2460 = \$30,483.41$
> This is an important point: Remember that the PV equation for a perpetuity (and an ordinary annuity) tells you the PV one period before the first cash flow. In this problem, since the cash flows are two years apart, we have found the value of the perpetuity one period (two years) before the first payment, which is one year ago. We need to compound this value for one year to find the value today. The value of the cash flows today is: $PV = \$30,483.41(1 + .11/365)^{365} = \$34,027.40$ The second part of the question assumes the perpetuity cash flows begin in four years. In this case, when we use the PV of a perpetuity equation, we find the value of the perpetuity two years from today. So, the value of these cash flows today is: $PV = \$30,483.41/(1 + .11/365)^{2(365)} = \$24,464.32$
> Context: An investment pays \$7,500 every other year forever. The discount rate is 11 percent compounded daily.
>
> Generated Question: What is the value of the investment if the first payment occurs one year from today? (Unit: dollar)
> Answer: 34027.40
>
> Generated Question: What is the value of the investment if the first payment occurs four year from today? (Unit: dollar)
> Answer: 24464.32

---

**Example 5 in prompt template for generating questions in financial calculation task.**

Original Question: An investment offers \$4,600 per year for 15 years, with the first payment occurring one year from now. If the required return is 8 percent, what is the value of the investment? What would the value be if the payments occurred for 40 years? For 75 years? Forever?

Original Answer: To find the PVA, we use the equation:
$PVA = C([1-[1/(1+r)]^t]/r)$
PVA@15 yrs: $PVA = \$4,600[[1-(1/1.08)^{15}]/.08] = \$39,373.60$
PVA@40 yrs: $PVA = \$4,600[[1-(1/1.08)^{40}]/.08] = \$54,853.22$
PVA@75 yrs: $PVA = \$4,600[[1-(1/1.08)^{75}]/.08] = \$57,320.99$
To find the PV of a perpetuity, we use the equation:
$PV = C/r$
$PV = \$4,600/.08 = \$57,500.00$
Notice that as the length of the annuity payments increases, the present value of the annuity approaches the present value of the perpetuity. The present value of the 75 year annuity and the present value of the perpetuity imply that the value today of all perpetuity payments beyond 75 years is only \$179.01.

Context: An investment offers \$4,600 per year for 15 years, with the first payment occurring one year from now. The required return is 8 percent

Generated Question: What is the value of the investment? (Unit: dollar)
Answer: 39373.60

Generated Question: If the payments occurred for 40 years, what is the value of the investment? (Unit: dollar)
Answer: 54853.22

Generated Question: If the payments occurred for 75 years, what is the value of the investment? (Unit: dollar)
Answer: 57320.99

Generated Question: If the payments occurred forever, what is the value of the investment? (Unit: dollar)
Answer: 57500.00

---

**Prompt template for verifying questions in *financial calculation* task.**

Original Question: {after-class question}
Original Answer: {after-class solution}
Context of Generated Question: {context}
Generated Question: {question}
Correct Answer: {answer}

Given the above original question and answer, please answer the following two questions.
Q1: Is the correct answer definitely true to the generated question?
Q2: Does the context provide the necessary information for the calculation to answer the generated question?

Your Answer to Q1 and Q2 (Yes or No, no explanation required):

### G.2.4 PROMPT FOR MATHEMATICAL CAPABILITY ANNOTATION

> **Prompt template for asking gpt-4o to annotate mathematical reasoning types to calculating questions.**
>
> Below are seven reasoning abilities required in solving math problems:
> 1. Numeric Commonsense: It involves intuitive understanding of daily numerical concepts, including understanding time differences, numerical judgment, and estimates. It covers temporal reasoning, spatial numeric assessments, and practical uses like budgeting and time reading.
> 2. Logical Reasoning: It focuses on critical thinking and deduction from provided information, including pattern recognition, sequence understanding, predictions, and statement evaluation. Key components include premises, conclusions, and the use of abstract reasoning.
> 3. Statistical Reasoning: It focuses on data interpretation and analysis, including measures (mean, median, mode), dispersion metrics (standard deviation, range), probability concepts, regression, correlation, and data inferences. It also identifies trends, outliers, and patterns.
> 4. Arithmetic Reasoning: It covers the fundamental operations such as addition, subtraction, multiplication, division, and understanding of number properties. It may also include the ability to interpret numerical data in different forms.
> 5. Algebraic Reasoning: It encompasses understanding variables, equations, and the manipulation of expressions with polynomials and exponents. It also covers solving simple to complex equations, and grasping functions, their properties, and graphical depictions.
> 6. Geometry Reasoning: It emphasizes spatial understanding, analysis of 2D and 3D figures, and reasoning about their shapes, sizes, and relationships. It includes symmetry, congruency, similarity, area, volume, and transformations.
> 7. Scientific Reasoning: It deals with the application of mathematical concepts in scientific contexts. This includes scientific notations, formula use, understanding rates, proportions, and percentages in practical situations, and problem-solving in scientific inquiries.
>
> Question: { question }
> Answer: { answer }
> Above is a calculating question along with its answer in finance domain. Plase label this question with at most two reasoning abilities defined above. You are NOT allowed to create other abilities. You should output your final answer with 'Therefore, my answer is'.
> Let's think step by step.

## G.3 PROMPT FOR EVALUATING BASELINES

Chain-of-thought prompt templates for evaluating baselines are shown in G.3.1. The program-of-thought prompt template for financial calculation task is shown in G.3.2.

### G.3.1 PROMPT FOR CHAIN-OF-THOUGHT METHOD

> **Prompt template for evaluation in *statement judging* task using CoT prompting.** `knowledge` is an empty string when no finance term is provided.
>
> {knowledge}
>
> Statement: {question}
> Is the above statement true or false? Please output your answer starting with 'Therefore, my answer is' at the end.
> Let's think step by step.

> Prompt template for evaluation in *multi-choice question answering* task using CoT prompting. `knowledge` is an empty string when no finance term is provided.
>
> {knowledge}
>
> Question: {question}
> Choices: {choices}
> Which one of the above choices is the most appropriate to answer the question? Please output your answer starting with 'Therefore, my answer is' at the end.
> Let's think step by step.

> Prompt template for evaluation in *financial calculation* task using CoT prompting. `knowledge` is an empty string when no finance term is provided.
>
> {knowledge}
>
> Question: {question}
> Please answer the above question and output your final answer starting with 'Therefore, my answer is' at the end, where you store you final answer into '[]'.
> Let's think step by step.

### G.3.2 PROMPT FOR PROGRAM-OF-THOUGHT METHOD

> Prompt template for evaluation in *financial calculation* task using PoT prompting. `knowledge` is an empty string when no finance term is provided.
>
> {knowledge}
>
> Question: {question}
> Please generate a Python program to answer the given question.
> '''python
> def solution():

## H ETHICS AND SOCIETAL IMPACT

We envision XFINBENCH as a comprehensive benchmark designed to assist researchers in evaluating the performance of their models within the finance domain. By offering a robust evaluation framework, XFINBENCH aims to drive advancements in foundational models for the research community, providing valuable insights into critical model capabilities such as temporal reasoning, future forecasting, scenario planning, numerical modeling, and cross-modal reasoning.

For constructing examples in XFINBENCH and finance terms for the knowledge bank, we primarily rely on textbooks that are openly available on the internet. Our annotators strictly adhere to copyright and licensing regulations, ensuring that data from sources prohibiting copying or redistribution is excluded. Furthermore, during the automated annotation and human quality validation processes for examples in XFINBENCH, we implement rigorous ethical guidelines to prevent biased content and safeguard against the inclusion of private data.

