# OpenReview forum: "FinBench: Benchmarking LLMs in Complex Financial Problem Solving and Reasoning"
_ICLR.cc/2025/Conference — ICLR 2025 Conference Withdrawn Submission_

### Official Review · Reviewer_ukQH · 2024-10-30

**Soundness:** 3
**Presentation:** 3
**Contribution:** 3
**Rating:** 5
**Confidence:** 3

**Summary:**

FINBENCH introduces a benchmark designed to evaluate LLMs' capabilities in complex financial problem-solving. The dataset comprises 4,235 examples sourced from graduate-level finance textbooks, assessing five core capabilities: terminology understanding, temporal reasoning, future forecasting, scenario planning, and numerical modeling. The evaluation is structured across three tasks: statement judging, multi-choice question answering, and financial calculation. Results indicate that even the best-performing model, o1, achieves an overall accuracy of 67.3%, which still falls significantly behind human experts by 12.5 percentage points.

**Strengths:**

First, the paper makes a significant contribution by introducing FINBENCH. It is the first comprehensive benchmark that extends beyond basic numerical calculations to evaluate advanced financial reasoning capabilities in LLMs. This represents a crucial step forward in understanding and improving AI systems' ability to handle complex financial tasks.

Second, the technical approach is robust and well-designed, featuring a systematic evaluation framework with five core capabilities and three distinct tasks. The methodology is strengthened by the inclusion of a substantial knowledge bank and the use of high-quality, graduate-level source material.

Third, the empirical evaluation is thorough and insightful, with comprehensive experiments across 18 leading models and detailed error analysis. The comparison with human expert performance provides valuable context for understanding current model limitations and sets clear benchmarks for future improvements in the field.

**Weaknesses:**

First, the paper lacks comprehensive comparisons with existing benchmarks in the financial domain. While specialized financial benchmarks may be limited, comparative analysis with recent benchmarks like MMLU-Pro and MMMU that include financial components would be valuable. A systematic comparison across dimensions such as task diversity, difficulty levels, and dataset size would help readers better understand FINBENCH's distinctive features and contributions to the field.

Second, the reliability of the ground truth labels raises concerns.  According to Section 2.3, the human expert sampling validation reveals an approximately 3% error rate in answer annotations, suggesting a potential uncertainty interval of around 6% in the leaderboard results.  Combined with the human expert accuracy of only 80%, the actual answer reliability could be even lower. The authors should consider conducting a complete manual verification of all 4,000+ questions to ensure benchmark quality and reliability.

Third, given that LLMs are highly sensitive to prompting, the evaluation would benefit from experiments with different prompt templates. Including ablation studies on various prompting strategies would provide deeper insights into FINBENCH's robustness and help readers better understand how the benchmark performs under different evaluation conditions. This additional analysis would strengthen the benchmark's validity and usefulness for the research community.

**Questions:**

1. Comparative Analysis:
How does FINBENCH compare to existing financial components in benchmarks like MMLU-Pro and MMMU? Could the authors provide a systematic comparison across dimensions such as task diversity, difficulty levels, and dataset size? This comparison would help readers better understand FINBENCH's unique contributions and positioning in the field of financial AI evaluation.

2. Label Reliability:
Given that human expert sampling revealed a 3% error rate in answer annotations (leading to a 6% uncertainty interval) and human expert accuracy is only 80%, what measures are being considered to improve the reliability of ground truth labels? Have the authors considered conducting a complete manual verification of all 4,000+ questions? How might this impact the current leaderboard results and benchmark reliability?

3. Prompt Sensitivity:
Considering the well-known sensitivity of LLMs to different prompting strategies, could the authors include experiments with various prompt templates? Specifically, how does FINBENCH performance vary across different prompting approaches? This analysis would provide valuable insights into the benchmark's robustness and help establish more reliable evaluation protocols.

---

### Official Review · Reviewer_CZE1 · 2024-11-02

**Soundness:** 3
**Presentation:** 3
**Contribution:** 1
**Rating:** 3
**Confidence:** 4

**Summary:**

The paper introduces a benchmark called "FINBENCH" to test large language models (LLMs) on complex finance tasks like understanding terminology, reasoning over time, forecasting, scenario planning, and numerical modeling. It highlights the gap between LLM performance and human experts, especially in handling calculations and multimodal data, and suggests that adding finance-specific knowledge can improve LLM accuracy.

**Strengths:**

- **Originality**: The dataset is sourced from financial textbooks, supplemented with enhancements from GPT-4o
- **Quality**: Human oversight in data collection and augmentation adds rigor, helping to ensure high data quality.
- **Clarity**: Visual materials, including images and tables, are clearly presented and effectively organized.
- **Significance**: This work offers a challenging multimodal benchmark for the financial domain

**Weaknesses:**

- Reliance on graduate-level textbooks as the only data source may limit diversity in data distribution. Have you considered some online resources, especially professional financial websites?
- No dataset or code provided

**Questions:**

- **Benchmark Comparisons**: Could you include performance results on other established benchmarks for the same models? This would help underscore the distinct challenges and complexity of your benchmark. (listed in Table 1)
- **Domain Comprehensiveness**: Given the specialized focus on finance, how does this benchmark ensure a comprehensive evaluation of a model’s financial knowledge, as well as its understanding and reasoning capabilities in this domain?
- **Data Leakage Prevention**: What specific measures has your team implemented to mitigate the risk of data leakage?

---

### Official Review · Reviewer_xUpo · 2024-11-04

**Soundness:** 4
**Presentation:** 3
**Contribution:** 3
**Rating:** 8
**Confidence:** 4

**Summary:**

This paper introduces FINBENCH, a benchmark specifically designed to evaluate large language models (LLMs) in handling complex, knowledge-intensive financial problems across a range of graduate-level finance topics with multimodal contexts.

It assesses five core capabilities critical to financial problem-solving: terminology understanding, temporal reasoning, future forecasting, scenario planning, and numerical modeling. The dataset, comprising 4,235 questions from textbooks and 3,032 finance terms, is divided into three tasks: statement judging, multiple-choice question answering, and financial calculation.

Experiments with 18 leading LLMs reveal that while the text-only model o1 achieves the highest accuracy among LLMs (67.3%), it remains significantly behind human expert performance, particularly in advanced reasoning tasks like temporal reasoning and scenario planning.

The authors also identify key performance challenges for LLMs, such as rounding errors in calculations and difficulties interpreting visual contexts.

**Strengths:**

1. Originality:  Unlike existing datasets that primarily focus on basic quantity extraction and mathematical reasoning, FINBENCH covers diverse graduate-level finance topics. This focus on complex, multimodal finance problems sets FINBENCH apart as an innovative contribution to the field.

2. Quality: The paper demonstrates thorough methodology and attention to detail in data collection, annotation, and validation.

3. Clarity: The paper is well-structured, with clear explanations of dataset construction, task classifications, evaluation metrics, and experimental setups.

4. Significance: FINBENCH’s insights into specific model limitations (e.g., rounding errors, visual-context challenges) are valuable for guiding future AI developments, especially for applications in finance and other high-stakes fields.

**Weaknesses:**

1. While FINBENCH is based on three classic finance textbooks, its range of topics may still lack full coverage of the broader finance domain. Expanding the dataset to include questions from additional, specialized finance areas would increase the benchmark’s applicability.

2. Although FINBENCH distinguishes itself from existing finance-related datasets, the paper could strengthen its contributions by running some of the same models on related benchmarks (e.g., TAT-QA, FinQA) and comparing performance directly.

**Questions:**

1. Do the authors plan to extend FINBENCH to include data with more diverse financial contexts or emerging topics?

2. Could the authors provide further insights on the inconsistent improvement of advanced capabilities through knowledge augmentation?

3. What specific steps could be taken to address common errors such as "blindness" in visual-context tasks?

---

### Official Review · Reviewer_sBF6 · 2024-11-05

**Soundness:** 2
**Presentation:** 2
**Contribution:** 2
**Rating:** 3
**Confidence:** 5

**Summary:**

The paper introduces FinBench, a specialized benchmark designed to evaluate the capabilities of LLMs in handling complex, knowledge-intensive financial problem-solving tasks. The benchmark assesses LLMs across five core capabilities: terminology understanding, temporal reasoning, future forecasting, scenario planning, and numerical modeling. It consists of 4,235 graduate-level finance examples spanning statement judging, multiple-choice question answering, and financial calculations. It also includes a knowledge bank of 3,032 finance terms to support model performance through knowledge augmentation.

**Strengths:**

- The paper is well-structured and easy to follow
- The proposed benchmark could be valuable to the research community
- The experiments are comprehensive, including many recently-relased LLMs

**Weaknesses:**

- The proposed benchmark heavily depends on GPT models for re-annotation and validation, which raises concerns about reliability and accuracy. Based on the prompts provided by the authors in the appendix, this process appears to involve numerous prompt-engineering techniques, which may introduce potential biases or inconsistencies in the generated problems. Although the authors report the human validation results of 1,000 examples. It is strongly encouraged to validate and correct each example by human expert evaluators.
- The implementation details of the knowledge augmentation setup lack clarity in both the main text and the appendix. Specifically, it is unclear how many retrieved knowledge pieces are incorporated into the demonstration and what specific prompt is used. Providing these details would enhance understanding.
- I have several concerns about the quality of the benchmark and the experimental setup, which are listed in the Questions section. Given the absence of provided code and data, my evaluation relies entirely on the content of the paper, limiting the depth of my assessment. I strongly encourage the authors to share at least the dataset, allowing reviewers to verify the paper.

**Questions:**

- The table caption should be placed above the table content according to the ICLR format instruction
- The term "finance subject" (e.g., in Figure 3) is not ideal; consider using "finance concept" instead, which more accurately conveys the intended meaning.
- Section 2.1, "Collection of Knowledge Bank", lacks sufficient detail, making it challenging for readers to fully understand the methodology. For example, the phrase "We then manually extract the definition of each term from the corresponding pages" raises questions: Who are "we"? What process is used to extract each term's definition? Are the definitions text-only, or do they also include mathematical expressions or figures? Providing specific examples in the Appendix would help clarify the knowledge bank collection process for readers.
- What are the qualifications and backgrounds of the annotators and evaluators? Are they experienced financial professionals, or are they the authors themselves?
- Recent studies indicate that Chain-of-Thought (CoT) reasoning may not consistently enhance model performance on reasoning-intensive tasks [1]. This raises the question: how does CoT reasoning perform specifically within the context of FinBench, compared to directly answering the question.
- Consider renaming the benchmark to differentiate it from existing works named FinBench [2-3]. This will help prevent confusion among future researchers and create a distinct identity for your contribution.



[1] Sprague, Z., Yin, F., Rodriguez, J. D., Jiang, D., Wadhwa, M., Singhal, P., ... & Durrett, G. (2024). To CoT or not to CoT? Chain-of-thought helps mainly on math and symbolic reasoning. arXiv preprint arXiv:2409.12183.
[2] Qi, S., Lin, H., Guo, Z., Szárnyas, G., Tong, B., Zhou, Y., ... & Steer, B. (2023). The LDBC Financial Benchmark. arXiv preprint arXiv:2306.15975.
[3] FinPT: Financial Risk Prediction with Profile Tuning on Pretrained Foundation Models

**Details Of Ethics Concerns:**

The paper uses problems collected from textbooks as examples, which may raise copyright concerns. However, the authors do not address these ethical considerations.

---

> ### Comment · Reviewer_sBF6 · 2024-11-28
>
> Thank you for your response. While I appreciate the effort put into addressing the feedback, I remain concerned about the quality of the benchmark. I recommend conducting another round of thorough validation before publication. Upon sampling examples from the GitHub repository provided, I identified multiple errors and instances of low quality in both the questions and solutions.
>
> I also reviewed the knowledge base and noticed similar issues. For instance, the first term in the knowledge base is listed as "external uses, using information from, Financial statements," which appears unclear and potentially misleading. This example raises concerns about the overall reliability of the knowledge base.
>
> Additionally, I strongly suggest the inclusion of detailed solutions for each example within the dataset, rather than providing only the ground-truth answers. This enhancement would greatly benefit future researchers by facilitating deeper and more accurate analysis of the dataset.
>
> Given these observations, I am inclined to maintain my initial score. I hope the authors take these suggestions into consideration to improve the overall quality and usability of the benchmark for future version

---

### Note · Authors · 2024-12-13

**Comment:**

We sincerely thank all reviewers and area chairs for their insightful feedback and thoughtful evaluation of our submission. After careful consideration, we have decided to withdraw our ICLR 2025 submission. We extend our best wishes to every one of you!

**Withdrawal Confirmation:**

I have read and agree with the venue's withdrawal policy on behalf of myself and my co-authors.